# BridgeVLA: Input-Output Alignment for Efficient 3D Manipulation Learning with Vision-Language Models

**Peiyan Li**[1,2,3,*], **Yixiang Chen**[1,3], **Hongtao Wu**[2,*,†], **Xiao Ma**[2,*], **Xiangnan Wu**[1]
**Yan Huang**[1,3,4,†], **Liang Wang**[1,3], **Tao Kong**[2], **Tieniu Tan**[1,3,5]
[1]New Laboratory of Pattern Recognition (NLPR),
Institute of Automation, Chinese Academy of Sciences
[2] ByteDance Seed [3]School of Artificial Intelligence, University of Chinese Academy of Sciences
[4] FiveAges [5] Nanjing University

## Abstract

Recently, leveraging pre-trained vision-language models (VLMs) for building vision-language-action (VLA) models has emerged as a promising approach to effective robot manipulation learning. However, only few methods incorporate 3D signals into VLMs for action prediction, and they do not fully leverage the spatial structure inherent in 3D data, leading to low data efficiency. In this paper, we introduce a new paradigm for constructing 3D VLAs. Specifically, we first pre-train the VLM backbone to take 2D images as input and produce 2D heatmaps as output. Using this pre-trained VLM as the backbone, we then fine-tune the entire VLA model while maintaining alignment between inputs and outputs by: (1) projecting raw point cloud inputs into multi-view images, and (2) predicting heatmaps before generating the final action. Extensive experiments show that the resulting model, BridgeVLA, can learn 3D manipulation both efficiently and effectively. BridgeVLA outperforms state-of-the-art baselines across three simulation benchmarks. In RLBench, it improves the average success rate from 81.4% to 88.2%. In COLOSSEUM, it demonstrates significantly better performance in challenging generalization settings, boosting the average success rate from 56.7% to 64.0%. In GemBench, it surpasses all the comparing baseline methods in terms of average success rate. In real-robot experiments, BridgeVLA outperforms a state-of-the-art baseline method by 32% on average. It generalizes robustly in multiple out-of-distribution settings, including visual disturbances and unseen instructions. Remarkably, it is able to achieve a success rate of 95.4% on 10+ tasks with only 3 trajectories per task, while other VLA methods such as $\pi_0$ fail completely. Project Website: `https://bridgevla.github.io/`.

## 1 Introduction

Leveraging pre-trained vision-language models (VLMs) [3, 43, 2, 24] for developing large vision-language-action (VLA) models has become a promising method for learning generalizable and robust manipulation policies [26, 4, 17, 31, 7]. However, most VLA models only incorporate 2D image inputs and require extensive efforts on data collection. On the other hand, 3D robot policies leverage 3D structural priors in model design and demonstrate exceptional sample efficiency in learning complex 3D robot manipulation tasks [39, 25, 13–15]. Can we develop a unified 3D VLA model which combines the effectiveness of VLA models with the efficiency from 3D policies?

---

[*]Project lead
[†]Corresponding author

39th Conference on Neural Information Processing Systems (NeurIPS 2025).

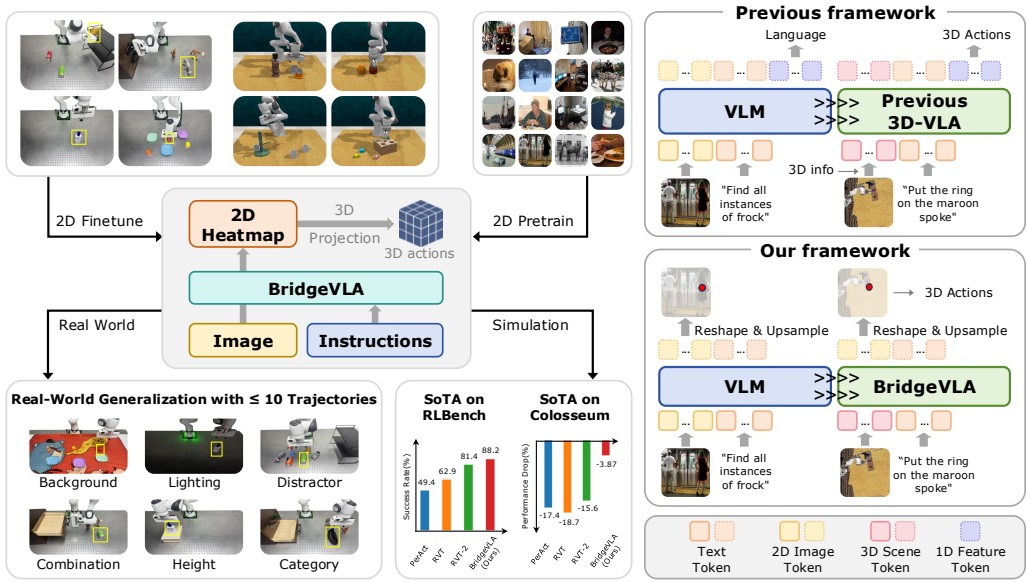

Figure 1: **Overview.** BridgeVLA is a novel 3D VLA model that aligns the input and output within a unified 2D image space. It is pre-trained on object grounding using 2D heatmaps and fine-tuned on action prediction for 3D manipulation. Experiment results in both simulation and the real world show that it is able to learn 3D manipulation both efficiently and effectively.

Although there have been some works exploring integrating 3D information into VLMs for developing 3D VLA models [52, 37], these works typically convert actions into token sequences that do not have spatial structure and use next-token prediction to predict actions. This strategy fails to take advantage of the 3D structural priors as previous efficient 3D policies [39, 25, 13–15] that align the observation input and action output into a unified space, therefore leading to poor sample efficiency. Another significant challenge in developing 3D VLA models lies in the misalignment between the 3D inputs used in action fine-tuning and the 2D image inputs used in original VLM pre-training, causing a large distributional shift from the original VLM pre-training.

To tackle the challenges mentioned above, as inllustrated in Fig. 1, we present BridgeVLA, a novel 3D VLA model that achieves remarkable sample efficiency and strong generalization capabilities. To ensure input alignment with the pre-trained VLM backbone, BridgeVLA transforms a 3D point cloud observation into multiple 2D images captured from different orthographic projection views [14, 15]. To leverage the structural priors of the 3D input, BridgeVLA is trained to predict 2D heatmaps for translational action prediction. The 2D heatmaps, generated from the tokens corresponding to the projection images, share the same resolution as these images, aligning the input observations and output actions within a unified spatial structure. Given that the original VLM is pre-trained to predict token sequences, which is incompatible with our VLA's 2D heatmap output, we also introduce a scalable pre-training method, which trains the model to ground objects with heatmaps conditioned on text inputs. This pre-training method equips the VLM with the capabilities to predict heatmaps before downstream fine-tuning for policy learning. **Overall, our design aligns the input and output within a shared 2D space in both pre-training and fine-tuning.**

We perform extensive experiments in both simulation and the real world to evaluate the proposed method. Results show that BridgeVLA is able to learn 3D manipulation both efficiently and effectively. It outperforms state-of-the-art baseline methods in RLBench [19], improving the average success rate from 81.4% to 88.2%. In COLOSSEUM [35], it showcases strong performance in challenging generalization settings, boosting the success rate from 56.7% to 64.0%. In GemBench [12], it surpasses all the comparing baseline methods in terms of average success rate. In real-robot experiments, we evaluate on seven different settings, spanning from visual perturbations to manipulating objects from unseen categories. BridgeVLA surpasses a state-of-the-art method by 32% on average and demonstrates strong performance in generalizing to multiple out-of-distribution settings. Notably, BridgeVLA is able to achieve a success rate of 96.8% on 10+ tasks using only 3 trajectories per task

for training, highlighting its superb sample efficiency. In summary, the contributions of this paper are threefold:

- We introduce BridgeVLA, a novel 3D VLA model that efficiently and effectively learns 3D robot manipulation with a vision-language model via input-output alignment with 2D heatmaps.
- We propose a scalable pre-training method to equip the model with the capability to predict heatmaps conditioned on text inputs via object grounding.
- We conduct extensive experiments in both simulation and real-world environments to thoroughly evaluate the proposed method. Results show that BridgeVLA outperforms state-of-the-art methods in both settings and achieves exceptional sample efficiency in real-robot experiments.

## 2 Related Work

**Language-Conditioned Visuomotor Policies.**   Most language-conditioned visuomotor policies employ transformers to process 2D visual inputs and directly generate 3D actions for manipulation [6, 7, 26, 4, 17, 10, 31, 30, 8, 51, 44]. In these works, leveraging pre-trained vision-language models (VLMs) for developing large vision-language-action (VLA) models has become popular for its effectiveness on learning complex manipulation [7, 26, 31, 4, 17]. However, such 2D image-based policies typically require significant efforts on data collection, often needing hundreds of trajectories per task to learn effectively. On the other hand, 3D manipulation policies hold great potential for efficient learning by taking advantage of the spatial structure inherent in the 3D inputs. A popular line of works take as inputs point cloud data [9, 48, 47, 13, 25]. For example, Act3D [13] proposes to create a 3D feature cloud by lifting image features to the observation point cloud and predicts translational actions via classification for 3D points in the observation space. Another line of works utilize voxels to represent the observation space and predict translational actions within the voxel space, unifying the input observation and output actions within the same space [39, 20]. Recently, RVT [14] and RVT-2 [15] propose to leverage orthographic projection of 3D point clouds to convert 3D signals to 2D images to avoid high computational cost on processing 3D inputs. Different from the above methods, our method aims to unify the effectiveness of VLA models and the efficiency of 3D policies within a single cohesive framework, combining the best of both worlds.

**3D Vision-Language-Action (VLA) Models.**   While 2D VLA models have been extensively studied, 3D VLA models [52, 22, 47, 29] remain relatively under-explored. Zhen *et al.* [52] build 3D-VLA on top of a large language model (LLM) and train the model to perform 3D reasoning, multi-modal goal generation, and robot planning. Lift3D [22] proposes to enhance 2D foundation models (*e.g.*, DINOv2 [34]) with implicit and explicit 3D robotic representation for learning 3D manipulation policies. FP3 [47] leverages a transformer to fuse the information from point clouds, proprioceptive states, and language instructions. PointVLA [29] utilizes a VLM and a point cloud encoder to process 2D images and 3D point clouds, respectively. The embeddings from both encoders are injected into an action expert for action prediction. SpatialVLA [37] introduces Ego3D position encoding to inject 3D information into 2D image observation and adaptive action grids to represent robot movement in a more transferable way. Our method is different from the above methods in that it is designed in a way to take advantage of the spatial structure of 3D inputs in action prediction. In addition, it bridges the gap between the 2D image inputs of pre-trained VLMs and the 3D inputs by projecting the 3D inputs into multiple 2D images instead of injecting 3D information into the VLMs. Such design enables it to simultaneously leverages the broad knowledge in the VLM backbone and the spatial structure priors embedded in 3D inputs.

## 3 BridgeVLA

### 3.1 Preliminaries

BridgeVLA aims to learn a multi-task 3D robot manipulation policy $\pi$, which maps the observation $\mathbf{o}$ and a language instruction $l$ to an action $\mathbf{a}$:

$$\pi : (\mathbf{o}, l) \mapsto \mathbf{a} \tag{1}$$

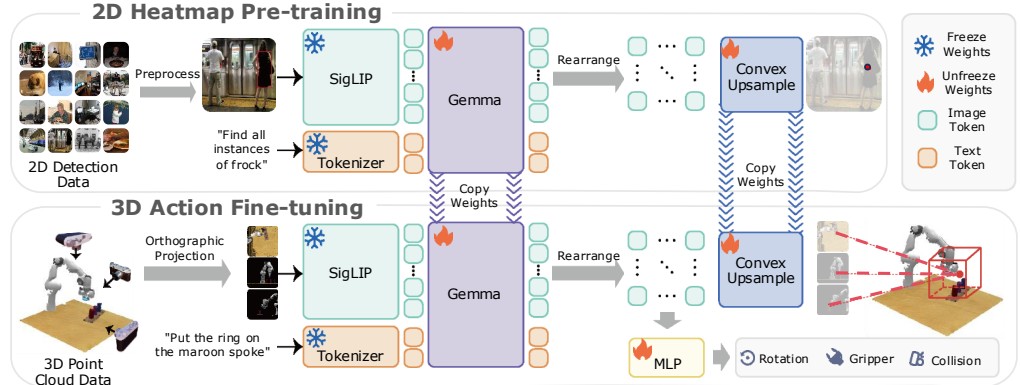

Figure 2: **Model Architecture.** (a) **2D Heatmap Pre-training:** we train BridgeVLA on 2D object detection datasets. The model takes as inputs an image and a language describing the target object and outputs a 2D heatmap which highlights regions of interest that correspond to the target object. Note that the bounding box shown here is for illustrative purposes only; it is not present in the image when input to the model. (b) **3D Action Fine-tuning:** the model takes as inputs three orthographic projection images of a 3D point cloud and a language instruction. It outputs three 2D heatmaps, which highlight the position of the end-effector in the next keyframe across all three views. For the remaining action components, it uses an MLP to process the image feature tokens to predict the rotation action, gripper action, and collision flag of the next keyframe.

We assume access to a set of expert demonstrations $\mathcal{D} = \{\tau^i\}_{i=1}^N$ containing $N$ trajectories. And each trajectory contains a language instruction and a sequence of observation-action pairs, *i.e.*, $\tau^i = \{l^i, (\mathbf{o}_1^i, \mathbf{a}_1^i), ..., (\mathbf{o}_H^i, \mathbf{a}_H^i)\}$. The observation $\mathbf{o}$ is one or multiple RGB-D images captured from one or multiple viewpoints. Following prior works [39, 14, 13], the action $\mathbf{a}$ consists of a 6-DoF end-effector pose $T \in SE(3)$, a target gripper state $g \in \{0, 1\}$, and a collision flag $c \in \{0, 1\}$ of the next key frame. The collision flag $c$ indicates whether the motion planner should avoid collisions while moving towards the target pose. A key frame typically captures important or bottleneck steps in a trajectory (detailed in appendix B.3) [23]. BridgeVLA operates through an iterative process: 1) predicting the action $\mathbf{a}_t$ conditioned on the current observation $\mathbf{o}_t$ and instruction $l$, 2) moving to the predicted next keyframe pose $T_t$ using a sampling-based motion planner [40, 28, 11], 3) updating observation and repeating until task completion or reaching a maximum step $H_{\max}$.

As illustrated in Fig. 2, BridgeVLA employs a dual-phase training recipe. During pre-training, it is trained to predict 2D heatmaps on object detection datasets. During fine-tuning, point clouds are projected into multiple 2D images as inputs to the VLM backbone. The model is trained to predict 2D heatmaps for estimating the translational action and other action components. **This design aligns the input and output within a shared 2D space in both pre-training and fine-tuning.**

### 3.2 2D-Heatmap Pre-training

The VLM backbone was originally pre-trained to predict token sequences without spatial structure. To equip it with the same ability to predict heatmaps as downstream policy learning, we introduce a pre-training stage which trains the model to ground target objects via heatmaps. Concretely, we leverage the 120K object detection split of RoboPoint [49] as our pre-training dataset. For each image, we construct the ground-truth heatmap $H^{\mathrm{gt}}$ from the bounding boxes of all objects of interest. Specifically, for each object, we construct a probability map with spatial truncation:

$$H_i^{\mathrm{gt}}(\mathbf{x}) = \begin{cases} p_i(\mathbf{x}) & \text{if } p_i(\mathbf{x}) \geq p_{\min} \\ 0 & \text{otherwise} \end{cases} \qquad (2)$$

where $\mathbf{x} = (u, v)$ denotes the pixel position, $p_i(\mathbf{x}) = \exp\left(-\|\mathbf{x} - \widehat{\mathbf{x}}_i\|^2 / 2\sigma^2\right)$, $\widehat{\mathbf{x}}_i$ is the center of the object bounding box, and $p_{\min}$ is a probability threshold. For all the objects of interest, we fuse the

probability map of all objects via averaging and normalization to obtain $H^{\text{gt}}$:

$$H^{\text{gt}}(\mathbf{x}) = \frac{H_{\text{avg}}(\mathbf{x})}{\sum_{\mathbf{x} \in \Omega} H_{\text{avg}}(\mathbf{x})}, \text{ where } H_{\text{avg}}(\mathbf{x}) = \frac{1}{N} \sum_{i=1}^{N} H_i^{\text{gt}}(\mathbf{x}) \tag{3}$$

where $\Omega$ denotes the pixel space. Please refer to Fig. 9 for samples of the ground-truth heatmaps.

As illustrated in Fig. 2, we input an image along with the text prompt describing the objects of interest into the VLM backbone of BridgeVLA. In this paper, we employ PaliGemma [3] as the VLM backbone, which consists of a SigLIP vision encoder [50] and a Gemma transformer backbone [41]. During its pre-training, PaliGemma takes as input one or multiple 2D images together with a prefix text (*e.g.*, a question about the image) and outputs a suffix text (*e.g.*, an answer to the question). While the model uses causal attention for predicting suffix text tokens, it adopts bidirectional attention for the image tokens and the prefix text tokens. This allows the image tokens to fuse information from the prefix text.

To predict the heatmap, we first rearrange the output image tokens according to their patch positions to reconstruct the spatial feature grid. A convex upsampling block [42] then converts the grid into a heatmap with the same resolution as the input image. Unlike fixed methods (e.g., bilinear or nearest-neighbor), this upsampling module learns pixel-wise interpolation weights, allowing for finer spatial detail recovery. The whole pipeline is trained with a cross-entropy loss to predict heatmaps that localize the position of all objects of interest in the image. We emphasize that the proposed pre-training strategy outputs a spatially aware 2D heatmap, in contrast to the conventional next-token-prediction used in prior works [52, 37]. Moreover, this approach is highly scalable, as it can, in principle, leverage any vision-language datasets that can be formulated as a heatmap prediction tasks, such as keypoint detection and semantic segmentation.

### 3.3 3D Action Fine-tuning

During fine-tuning, we first reconstruct a point cloud of the scene from the RGB-D images captured from calibrated cameras. To align with the 2D image input of the VLM backbone, we render three orthographic projection images of the point cloud from three viewpoints (top, front, and right) and use these images as the input images for the VLM backbone as in RVT [14] and RVT-2 [15]. These images, along with the task instruction, are then fed into the pre-trained VLM backbone to generate a heatmap for each of the three views. Importantly, we do not incorporate any additional information (*e.g.*, robot states) during the VLM forward pass to minimize the distribution shift between pre-training and fine-tuning.

For translational actions, we back-project the heatmaps of all three views to estimate the scores of all 3D point grids distributed uniformly across the robot workspace. The position of the 3D point with the highest score determines the end-effector's translation in the next keyframe. Similar to previous works [14, 15], we use Euler angles to represent rotational actions where each axis is discretized into 72 bins. To predict the rotation, binary gripper action, and collision avoidance flag, we integrate features from global and local contexts. For the global feature, max-pooling is applied to the output tokens of each inputted orthographic projection image, resulting in three tokens in total – one for each view. For the local feature, we extract a token from the heatmap peak of each view, also resulting in three tokens in total. All these tokens are concatenated and passed through MLP to predict the rotation action, gripper action, and collision avoidance flag.

Following the approach in prior works [20, 15], BridgeVLA adopts a coarse-to-fine refinement strategy for accurate action prediction. After the initial prediction on the original point cloud, we zoom in and crop the point cloud with a cuboid centered at the predicted translation. A second forward pass is performed on the cropped, zoomed-in point cloud. The predicted action from the second pass is used for execution.

The training loss during fine-tuning consists of four components:

$$L = L_{\text{trans}} + L_{\text{rot}} + L_{\text{gripper}} + L_{\text{collision}} \tag{4}$$

Similar to pre-training, $L_{\text{trans}}$ is a cross-entropy loss that supervises the heatmap prediction for translational actions. The ground-truth heatmap for each orthographic view is the normalized single-object probability map defined in Eq. 2, where $\widehat{\mathbf{x}}_i$ represents the projected pixel position of the ground-truth end-effector position in the next keyframe. As we discretize the Euler angles for rotation

| Models | Overall | | Task Success Rate (%) | | | | | | | |
|---|---|---|---|---|---|---|---|---|---|---|
| | Avg. SR (%)↑ | Avg. Rank↓ | Close Jar | Drag Stick | Insert Peg | Meat off Grill | Open Drawer | Place Cups | Place Wine | Push Buttons |
| Image-BC (CNN) [21, 39] | 1.3 | 11.72 | 0.0 | 0.0 | 0.0 | 0.0 | 0.0 | 4.0 | 0.0 | 0.0 |
| Image-BC (ViT) [21, 39] | 1.3 | 12.19 | 0.0 | 0.0 | 0.0 | 0.0 | 0.0 | 0.0 | 0.0 | 0.0 |
| C2F-ARM-BC [20, 39] | 20.1 | 10.72 | 24.0 | 24.0 | 4.0 | 20.0 | 20.0 | 0.0 | 8.0 | 72.0 |
| HiveFormer [16] | 45.3 | 8.47 | 52.0 | 76.0 | 0.0 | **100.0** | 52.0 | 0.0 | 80.0 | 84.0 |
| PolarNet [9] | 46.4 | 7.61 | 36.0 | 92.0 | 4.0 | **100.0** | 84.0 | 0.0 | 40.0 | 96.0 |
| PerAct [18] | 49.4 | 7.0 | 55.2±4.7 | 89.6±4.1 | 5.6±4.1 | 70.4±2.0 | 88.0±5.7 | 2.4±3.2 | 44.8±7.8 | 92.8±3.0 |
| Act3D [13] | 65.0 | 4.89 | 92.0 | 92.0 | 27.0 | 94.0 | 93.0 | 3.0 | 80.0 | 99.0 |
| RVT [14] | 62.9 | 4.92 | 52.0±2.5 | 99.2±1.6 | 11.2±3.0 | 88.0±2.5 | 71.2±6.9 | 4.0±2.5 | 91.0±5.2 | **100.0±0.0** |
| 3D Diffuser Actor [25] | 81.3 | 2.67 | 96.0±2.5 | **100.0±0.0** | 65.6±4.1 | 96.8±1.6 | 89.6±4.1 | 24.0±7.6 | 93.6±4.8 | 98.4±2.0 |
| RVT-2 [15] | 81.4 | 2.75 | **100.0±0.0** | 99.0±1.7 | 40.0±0.0 | 99.0±1.7 | 74.0±11.8 | 38.0±4.5 | **95.0±3.3** | **100.0±0.0** |
| BridgeVLA w/o heat | 31.4 | 10.06 | 49.3±2.3 | 65.3±2.3 | 0.0±0.0 | 81.3±4.6 | 74.7±10.1 | 1.3±2.3 | 32.0±14.4 | 54.7±6.1 |
| BridgeVLA w pos | 56.2 | 5.97 | 96.0±0.0 | 58.7±6.1 | 26.7±2.3 | 96.0±0.0 | 97.3±2.3 | 14.7±4.6 | 81.3±8.3 | 86.7±2.3 |
| **BridgeVLA** | **88.2** | **2.03** | **100.0±0.0** | **100.0±0.0** | **88.0±2.8** | **100.0±0.0** | **100.0±0.0** | **58.4±10.0** | 88.0±2.8 | 98.4±2.2 |
| Models | Put in Cupboard | Put in Drawer | Put in Safe | Screw Bulb | Slide Block | Sort Shape | Stack Blocks | Stack Cups | Sweep to Dustpan | Turn Tap |
| Image-BC (CNN) [21, 39] | 0.0 | 8.0 | 4.0 | 0.0 | 0.0 | 0.0 | 0.0 | 0.0 | 0.0 | 8.0 |
| Image-BC (ViT) [21, 39] | 0.0 | 0.0 | 0.0 | 0.0 | 0.0 | 0.0 | 0.0 | 0.0 | 0.0 | 16.0 |
| C2F-ARM-BC [20, 39] | 0.0 | 4.0 | 12.0 | 8.0 | 16.0 | 8.0 | 0.0 | 0.0 | 0.0 | 68.0 |
| HiveFormer [16] | 32.0 | 68.0 | 76.0 | 8.0 | 64.0 | 8.0 | 8.0 | 0.0 | 28.0 | 80.0 |
| PolarNet [9] | 12.0 | 32.0 | 84.0 | 44.0 | 56.0 | 12.0 | 4.0 | 8.0 | 52.0 | 80.0 |
| PerAct [18] | 28.0±4.4 | 51.2±4.7 | 84.0±3.6 | 17.6±2.0 | 74.0±13.0 | 16.8±4.7 | 26.4±3.2 | 2.4±2.0 | 52.0±0.0 | 88.0±4.4 |
| Act3D [13] | 51.0 | 90.0 | 95.0 | 47.0 | 93.0 | 8.0 | 12.0 | 9.0 | 92.0 | 94.0 |
| RVT [14] | 49.6±3.2 | 88.0±5.7 | 91.2±3.0 | 48.0±5.7 | 81.6±5.4 | 36.0±2.5 | 28.8±3.9 | 26.4±8.2 | 72.0±0.0 | 93.6±4.1 |
| 3D Diffuser Actor [25] | **85.6±4.1** | 96.0±3.6 | 97.6±2.0 | 82.4±2.0 | **97.6±3.2** | 44.0±4.4 | 68.3±3.3 | 47.2±8.5 | 84.0±4.4 | **99.2±1.6** |
| RVT-2 [15] | 66.0±4.5 | 96.0±0.0 | 96.0±2.8 | **88.0±4.9** | 92.0±2.8 | 35.0±7.1 | **80.0±2.8** | 69.0±5.9 | **100.0±0.0** | 99.0±1.7 |
| BridgeVLA w/o heat | 5.3±2.3 | 0.0±0.0 | 58.7±22.7 | 2.7±2.3 | 64.0±0.0 | 4.0±4.0 | 0.0±0.0 | 0.0±0.0 | 32.0±4.0 | 40.0±10.6 |
| BridgeVLA w pos | 10.7±2.3 | 78.7±2.3 | 97.3±4.6 | 16.0±4.0 | 72.0±0.0 | 21.3±8.3 | 17.3±2.3 | 4.0±4.0 | 53.3±2.3 | 84.0±0.0 |
| **BridgeVLA** | 73.6±4.6 | **99.2±1.8** | **99.2±1.8** | 87.2±6.6 | 96.0±2.8 | **60.8±7.7** | 76.8±8.7 | **81.6±3.6** | 87.2±1.8 | 92.8±3.3 |

Table 1: **Results on RLBench.** The "Avg. Rank" column reports the average rank of each method across all 18 tasks, where lower values indicate better overall performance. "BridgeVLA w/o heat" refers to the ablated version that directly predicts actions without using intermediate heatmaps. "BridgeVLA w pos" refers to the ablated version that incorporates position features into the image features. BridgeVLA achieves the best performance in 10 out of the 18 tasks.

into bins, we also apply cross-entropy loss in $L_{\text{rot}}$ to supervise rotation prediction. For gripper action and collision avoidance, we use the binary cross-entropy loss in $L_{\text{gripper}}$ and $L_{\text{collision}}$ as supervision. To enhance geometric robustness, random rigid-body transformations are applied jointly to the point cloud and the ground-truth action during training. Additional training details can be found in Appendix A.

## 4 Experiments

In this section, we perform extensive experiments in both simulation and the real world to evaluate the proposed method. Through the experiments, we aim to answer five questions:

- **Q1:** How effectively does BridgeVLA learn 3D robot manipulation compared to state-of-the-art methods when sufficient data is available?

- **Q2:** Does BridgeVLA learn more efficiently than existing state-of-the-art methods when data is limited (e.g., 3 trajectories per task)?

- **Q3:** How robust is BridgeVLA in handling visual disturbances (e.g., distractors, background, and lighting)?

- **Q4:** How well does BridgeVLA generalize to novel object-skill combinations and objects from previously unseen categories?

- **Q5:** Are our architectural designs (e.g., predicting heatmaps before outputting actions) truly useful when constructing 3D VLA?

### 4.1 Simulation Experiments

#### 4.1.1 Experiments on RLBench

**Setup.** RLBench [19] implements tasks in CoppeliaSim [38] using a Franka Panda robot mounted with a parallel-jaw gripper. The observation contains four RGB-D images captured from four calibrated cameras positioned at the front, left shoulder, right should, and wrist. Following previous works [39, 13, 14, 25, 15], we perform experiments on 18 tasks from RLBench. These tasks span 1)

non-prehensile manipulation (*e.g.*, *slide block to target*), 2) pick-and-place (*e.g.*, *stack cups*), and 3) high-precision insertion (*e.g.*, *insert peg*). Each task is provided with 100 expert demonstrations. And each demonstration is paired with language instruction and multiple keyframes. Models are evaluated via binary success rates over 25 trials per task, with a maximum of 25 action steps per trial.

**Baselines.** We compare BridgeVLA with multiple baselines. (1) **Image-BC (CNN)** and **Image-BC (ViT)** [21] are two 2D baseline methods which predict the actions directly from 2D images using CNN and ViT as the backbone, respectively. (2) **C2F-ARM-BC** [20] predicts the next keyframe action in the voxel space with a coarse-to-fine strategy. (3) **PerAct** [39] also operates in the voxel space and predicts the action with a perciever transformer [18]. (4) **HiveFormer** incorporates historical information using a unified multi-modal transformer architecture. (5) **PolarNet** employs PointNext [36] to encode the 3D scene and predicts both heatmaps and offsets for all points to estimate translational actions. (6) **Act3D** [13] predicts the next keyframe action by selecting the point with the highest score from a set of randomly sampled points in the workspace. (7) **3D Diffuser Actor** [25] generates 3D trajectories via a diffusion process conditioned on 3D observation and language instructions. (8) **RVT** [14] uses multi-view transformer to aggregate information from multiple orthographic views of the point cloud observation. (9) And **RVT-2** [15], the current state-of-the-art method, further improves the precision of its prior via a coarse-to-fine strategy.

**Results.** In total, we evaluate BridgeVLA five times to minimize statistical bias. The results are shown in Table 1. BridgeVLA outperforms all the comparing baseline methods, achieving an average success rate of 88.2% and an average rank of 1.9 across all the 18 tasks, establishing a new state of the art in this benchmark. These results address Q1, demonstrating the effectiveness of BridgeVLA in learning complex 3D manipulation tasks. We highlight that BridgeVLA outperforms the best baseline method by a large margin in *Insert Peg* (88.0% vs 40.0%) and *Sort Shape* (60.8% vs 35.0%). These two tasks demand highly precise alignment between the peg and hole and the block and sorter, respectively. The high success rates of our method showcase its strong capabilities of learning precise manipulation which is highly desirable in many industrial applications. Among the 18 tasks, BridgeVLA performs the worst in *Place Cups*, despite surpassing all the comparing baseline methods. We hypothesize this is because the target keypoints are often occluded in all orthographic projection views, which makes learning and prediction more challenging. In the future, we plan to explore dynamically selecting the projection views for rendering to avoid this problem.

### 4.1.2 Experiments on COLOSSEUM & GemBench

To further evaluate the generalization capabilities of BridgeVLA, we conduct experiments on the COLOSSEUM benchmark [35] and GemBench [12]. These two benchmarks extend RLBench. The COLOSSEUM benchmark evaluates models in environments with 12 axes of perturbations, which were not seen during training. These perturbations include variations in object texture, color, size, background, lighting, distractors, and camera poses. As such, this benchmark is used to assess Q3.

GemBench is a hierarchical generalization benchmark. Its training set consists of 16 tasks (31 variations) covering seven core action primitives: press, pick, push, screw, close, open, and stack/put. The test set includes 44 tasks (92 variations), categorized into four increasingly challenging settings. These settings incorporate novel object-skill combinations and new object categories, making it suitable for evaluating Q4.

BridgeVLA outperforms all existing state-of-the-art 3D manipulation methods on both benchmarks, addressing Q3 and Q4. Due to space limitations, the details of the environment setup, baselines, and analysis can be found in Appendix B.1 and Appendix B.2.

### 4.2 Real-Robot Experiments

**Setup.** In this section, we perform real-robot experiments to validate the effectiveness of BridgeVLA in the real world. Our real-robot setup includes a Franka Research 3 robot arm mounted with a parallel-jaw gripper (Fig. 3). A static ZED 2i depth camera is used to provide the colored point cloud observation. In total, we evaluate on 13 tasks (see Tab. 12 for a full list of tasks). These tasks ranges from simple pick-and-place to complex long-horizon tasks, requiring the robot to open a drawer and put items into the drawer. Each task contains 3-9 keyframes (see Fig. 7 and 8 for visualization). For each task, we collect 10 expert trajectories for training.

| Method | Put the soda can in the bottom shelf | Put the giraffe in the lower drawer | Place the red block in the blue plate | Press Sanitizer | Put the RedBull can in the top shelf | Put the RedBull can in the bottom shelf | Put the coke can in the top shelf |
|---|---|---|---|---|---|---|---|
| SpatialVLA(50) [37] | 1/10 | 1/10 | 5/10 | 6/10 | 3/10 | 1/10 | 2/10 |
| SpatialVLA(10) [37] | 0/10 | 0/10 | 0/10 | 2/10 | 0/10 | 0/10 | 0/10 |
| $\pi_0$ [5] | 0/10 | 0/10 | 2/10 | 1/10 | 0/10 | 1/10 | 0/10 |
| ACT [51] | 2/10 | 2/10 | 3/10 | 2/10 | 3/10 | 1/10 | 2/10 |
| RVT-2 [15] | 10/10 | 8/10 | 8/10 | 10/10 | 9/10 | 10/10 | 10/10 |
| BridgeVLA | 9/10 | 9/10 | 10/10 | 10/10 | 10/10 | 10/10 | 10/10 |

| Method | Place the orange block in the green plate | Place the red block in the purple plate | Place the yellow block in the green plate | Put the zebra in the upper drawer | Put the zebra in the lower drawer | Put the wolf in the upper drawer | Average |
|---|---|---|---|---|---|---|---|
| SpatialVLA(50) [37] | 6/10 | 3/10 | 5/10 | 2/10 | 0/10 | 2/10 | 28.5% |
| SpatialVLA(10) [37] | 1/10 | 1/10 | 0/10 | 0/10 | 0/10 | 0/10 | 3.1% |
| $\pi_0$ [5] | 0/10 | 0/10 | 1/10 | 0/10 | 0/10 | 0/10 | 3.8% |
| ACT [51] | 2/10 | 3/10 | 4/10 | 1/10 | 2/10 | 1/10 | 22.3% |
| RVT-2 [15] | 10/10 | 9/10 | 9/10 | 7/10 | 8/10 | 9/10 | 90% |
| BridgeVLA | 10/10 | 10/10 | 10/10 | 9/10 | 10/10 | 9/10 | 96.9% |

Table 2: **Per-task Success Rate in the Basic Setting.** Except for SpatialVLA(50), which was trained with 50 trajectories, all other methods were trained with 10 trajectories. BridgeVLA outperforms all baseline methods, achieving an almost perfect success rate of 96.9%.

In total, we design 7 different settings to comprehensively evaluate our model's performance. (1) **Basic**: The model is evaluated in environments that are similar to the training data. (2) **Distractor**: Distractor objects that are visually similar to at least one target object are added to the scene. (3) **Lighting**: The model is tested in a visually distinct lighting condition in which the lights are turned off. (4) **Background**: Three different tablecloths are used to change the background. (5) **Height**: All objects for manipulation are placed on a drawer that is 9.5cm high. (6) **Combination**: We combine objects and skills that are not paired together in the training datasets. That is, while the objects (*e.g.*, red block and green plate) and skill (*e.g.*, place A in B) are seen during training, the instruction that pairs them together is novel (*e.g.*, place the red block in the green plate). In total, we evaluate 13 novel object-skill combinations (Fig. 11 and 12). (7) **Category**: To test whether BridgeVLA is able to transfer the broad knowledge from pre-training to downstream policy learning, we evaluate on manipulating objects from categories that are *unseen* in the robot training data. In total, we test 7 novel objects (Fig. 13). Distractor, Lighting, Background, and Height aim to evaluate the robustness against visual disturbances, while Combination and Category evaluate the generalization capabilities for unseen instructions.

To demonstrate BridgeVLA's advantages over existing manipulation policy, we compare it with four types of representative methods:

1) **SpatialVLA** [37]: A state-of-the-art **3D VLA** model that incorporates 3D information through Ego3D positional encoding and leverages Adaptive Action Grids to accelerate inference.

2) $\pi_0$ [5]: A state-of-the-art **2D VLA** model pretrained on a large-scale cross-embodiment dataset. It adopts a vision-language model (VLM) backbone and employs a flow matching action expert to generate final actions.

3) **ACT** [51]: A state-of-the-art **2D non-VLA** model using a Conditional Variational Autoencoder (CVAE) to model action distributions. Though effective for fine-grained manipulation, ACT does not support language conditioning, so we train a separate single-task model for each task, which should theoretically perform better than a multi-task version.

4) **RVT-2** [15]: A state-of-the-art **3D non-VLA** model performing the best in our simulation experiments. (See Sec. 4.1)

**Results.** We first compare BridgeVLA with these baselines on the basic setting. For every task, we evaluated every baseline over 10 trials to ensure statistical robustness. For fair comparison, we photographed each test scene and manually aligned the scenes across all methods. Results are provided in Tab. 2. As we can see, most methods completely fails when given only 10 trajectories per task except two 3D related methods: RVT-2 and BridgeVLA. Notably, although SpatialVLA also utilises 3D information, its data efficiency is still very low. Even when the data are increased to 50 trajectories per task, its success rate is still much lower than BridgeVLA, which indicates only adding 3D information is not enough for constructing 3D VLA model and a carefully designed network architecture is still very important. To assess the data efficiency of BridgeVLA, we also train the model with only 3 trajectories per task. Remarkably, despite the limited data, BridgeVLA achieves a success rate of $95.4\%$ in Basic, matching the performance achieved with 10 trajectories per task. This result underscores the data efficiency of the proposed method, directly addressing Q2. Detailed

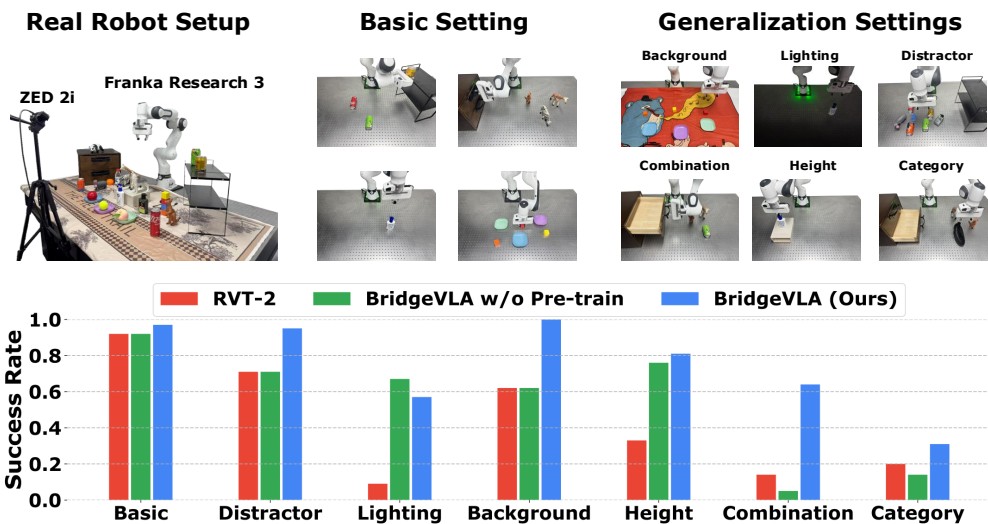

Figure 3: **Real-Robot Experiments and Results.** We use a Franka Research 3 robot arm and a ZED 2i camera to capture point clouds of the scene. To evaluate the model's performance, we design 7 different settings including one basic setting and six generalization settings. Experimental results show that BridgeVLA outperforms the state-of-the-art baseline method RVT-2 [15] by an average of 32%.

per-task results are provided in Appendix C.5 and observations about the baselines are detailed in Appendix C.2.

Considering that only RVT-2 and BridgeVLA have a good performance in basic setting, we only further evaluate these two methods on other generalization settings. The average success rates are shown in Fig. 3. BridgeVLA outperforms RVT-2 in all the seven settings. As we can see, RVT-2 struggles in both visual generalization settings and semantic generalization settings, while BridgeVLA performs much better, especially in Lighting and Combination. These results addresses Q3 and Q4, indicating that BridgeVLA is able to handle visual disturbance and novel instructions robustly.

Although our method outperforms baseline methods in the Category setting, its absolute success rate is not high. A common failure mode is that the robot often ignores the target object and moves directly to the destination during pick-and-place manipulation. We believe this relatively low performance is *not* due to BridgeVLA forgetting the knowledge gained from pre-training, as it still predicts heatmaps accurately when provided with samples from the pre-training dataset after fine-tuning (see Fig. 4 and Appendix C.4). Instead, we hypothesize that the reduced performance stems from two factors: 1) The images in the pre-training dataset are mostly captured from third-person views, which differ significantly from the projection images in our robot data; 2) The pre-training task focuses solely on object localization, whereas manipulation involves predicting keypoints that do not correspond to an object. To address these issues, we plan to expand both the scale and diversity of the pre-training dataset and explore more expressive action-decoding methods to better leverage the preserved pre-training knowledge.

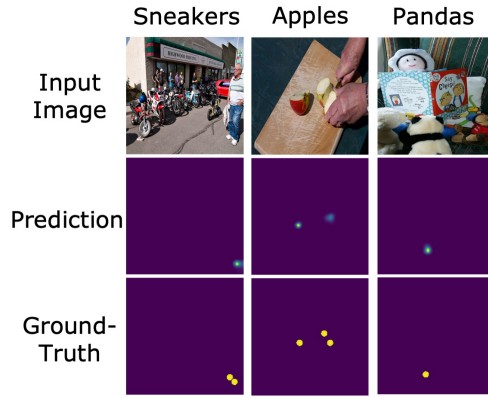

Figure 4: **Prediction on Pre-training Data after Fine-tuning.** To simulate the multi-view inputs during fine-tuning, we repeat each pre-training image three times and feed them into the fine-tuned model to generate heatmaps. Note that these samples are *not* cherry-picked. Additional samples can be found in Appendix C.4.

### 4.3 Ablation Studies

To prove the effectiveness of our model design and provide insights for the community, we conduct three ablation studies:

**Whether we need to predict heatmaps before predicting actions.** Our approach avoids direct action prediction by first generating 2D heatmaps using a convex upsampling module. Target positions are then computed by projecting 3D workspace points onto the heatmaps and selecting the point with the highest mean probability. For ablation, we replaced the convex upsampling module (309M parameters) with a similarly sized Transformer decoder (303M) to directly predict target positions, supervised by MSE loss. All other modules were kept the same as before. We performed a hyperparameter grid search and evaluated the model on RLBench. Results are shown in the Tab. 1. Replacing heatmap prediction with direct position regression reduced the average success rate from 88.2% to 31.4%, confirming the effectiveness of our heatmap-based design. The ablated model was also harder to train and more sensitive to hyperparameters—requiring a batch size of 192 and careful learning rate tuning—while our original model trains reliably even with a batch size of 64. We see three main reasons for this outcome: (1) Heatmaps offer denser supervision than 3D position vectors, enabling more effective learning. (2) Projecting 3D points onto heatmaps introduces helpful spatial priors, easing the learning process. (3) The 2D heatmaps share the same spatial structure as the input images, enhancing alignment and improving performance.

**Whether we need to remove the 3D position input to the VLM backbone.** Unlike typical 3D VLA models like SpatialVLA, we deliberately avoid using per-pixel 3D position inputs and rely solely on RGB images. This design preserves alignment between the input feature spaces of fine-tuning and VLM pretraining, which we find crucial for effective vision-language-action (VLA) modeling. To test this, we added a 3D convolutional module to encode per-pixel 3D positions, fused them with 2D features, and fed the result into the backbone. Although this adds richer spatial cues, it alters the image feature distribution seen during pretraining, leading to a performance drop from 88.2% to 56.2% on RLBench. Detailed results are shown in the Tab. 1.

**Whether we need to do 2D heatmap pre-training to the VLM backbone.** The original VLM backbone can not predict heatmaps, while our downstream policy learning requires such ability. To bridge the gap, we do 2D heatmap pre-training to the VLM backbone. To verify its effectiveness, we ablate this pre-training and evaluate model's performance in the real world, the results are shown in Fig. 3. As we can see, BridgeVLA w/o Pre-train is not able to generalize well in both language-related generalization settings and can not even beat RVT-2, while BridgeVLA achieves the best performance across the two generalization settings especially in Combination, highlighting its ability to understand language semantics. We hypothesize that the 2D-heatmap pre-training equips BridgeVLA with the ability to connect the semantics in language instructions with image observations in the heatmap space. All the above experiment results address Q5 and highlight the effectiveness of our architectural designs.

## 5   Conclusions & Future Work

This paper has introduced BridgeVLA, a novel and efficient 3D vision-language-action (VLA) model built on top of a pre-trained vision-language model (VLM) [3]. Keys to our method are that (1) it converts 3D inputs to 2D images to align with the 2D image inputs of the pre-trained VLM; (2) it aligns the input observation and the output action to a unified 2D image space via 2D heatmap prediction; (3) it adopts a scalable pre-training method to equip the VLM with the capability to predict heatmaps before fine-tuning on action prediction. Extensive experiments show that the proposed method is able to learn 3D manipulation efficiently and effectively in both simulation and the real world. In the future, we plan to explore pre-training on more diverse tasks, including semantic segmentation and keypoint detection. We also want to incorporate more expressive action-decoding methods (*e.g.*, diffusion [10]) into the framework to continue improving the policy performance.

## 6   Acknowledgments

This work was jointly supported by National Natural Science Foundation of China (62322607, 62236010 and 62276261), Beijing Natural Science Foundation (L252033), FiveAges Grant, and Youth Innovation Promotion Association of Chinese Academy of Sciences under Grant 2021128.

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

# —————Appendix—————

Table 3: Training hyperparameters for BridgeVLA

|  | Pretrain | RLBench Finetune | Colosseum Finetune | Real-robot Finetune |
|---|---|---|---|---|
| learning rate | 5e-5 | 8e-5 | 8e-5 | 2e-5 |
| optimizer | AdamW | AdamW | AdamW | AdamW |
| batch size | 384 | 192 | 192 | 192 |
| warmup steps | 400 | - | - | - |

## A   Training & Inference Details

Detailed training configurations are summarized in Tab. 3. Throughout both pre-training and fine-tuning, we keep the SigLIP vision encoder and language token embeddings frozen.

**Computational Resources:**

1. Pre-training: 8 NVIDIA A100 GPUs for 3,800 steps ($\approx$2 hours)

2. RLBench fine-tuning: 48 NVIDIA H100 GPUs for 83,000 steps ($\approx$20 hours)

3. COLOSSEUM fine-tuning: 48 NVIDIA H100 GPUs for 83,000 steps ($\approx$20 hours)

4. GemBench fine-tuning: 40 NVIDIA A100 GPUs for 50 epochs ($\approx$2.1 hours)

5. Real-world fine-tuning: 8 NVIDIA A100 GPUs for 300 epochs ($\approx$1.5 hours)

For inference, we run BridgeVLA on a machine equipped with an NVIDIA RTX 4090 GPU. To evaluate its inference speed, we conducted 100 trials. From point cloud input to action output, the average end-to-end inference time is 0.21 seconds.

## B   Simulation Experiments

### B.1   Experiments on COLOSSEUM

**Setup.**   The COLOSSEUM benchmark is an extension to the RLBench benchmark. The model is trained on the data from the original RLBench benchmark but evaluated in environments spanning 12 axes of perturbations. These perturbations, which are unseen during training, encompass changes in object texture, color, and size, backgrounds, lighting, distractors and camera poses. In total, the COLOSSEUM creates 20,371 unique task perturbations instances to comprehensively evaluate the generalization capabilities of the model. Specifically, our evaluation includes three steps: 1) train the model with the original RLBench data without perturbations (100 trajectories per task) on 20 tasks, 2) evaluate each task over 25 trials per perturbation, 3) compute the average success rate of all evaluated tasks for every perturbation. Besides the 12 types of perturbations, we also evaluate on basic variations from the original RLBench (denoted as **RLBench** in Tab. 4), and a more challenging setting which combines all the 12 types of perturbations (denoted as **All Perturbations** in Tab. 4).

**Baselines.**   We compare BridgeVLA with five baseline methods. **R3M-MLP** and **MVP-MLP** are two 2D methods that utilize pre-trained visual encoders to process observation images and an MLP for action prediction. Specifically, R3M-MLP uses R3M [33] that is pre-trained on large-scale egocentric human videos; MVP-MLP uses MVP [46] that is pre-trained on millions of in-the-wild data. Both visual encoders show strong adaptability on various robotics tasks in both simulation and the real world. We also compare with three 3D methods introduced in Sec. 4.1.1, *i.e.*, **PerAct** [39], **RVT** [14], and **RVT-2** [15].

| | Overall | | Success Rate (%) | | | | | |
|---|---|---|---|---|---|---|---|---|
| **Models** | **Avg. SR (%) ↑** | **Avg. Rank ↓** | All Perturbations | MO-COLOR | RO-COLOR | MO-TEXTURE | RO-TEXTURE | MO-SIZE |
| R3M-MLP[33] | 0.8 | 5.71 | 0.6 | 0.4 | 0.0 | 0.0 | 0.0 | 1.8 |
| MVP-MLP[46] | 1.6 | 5.0 | 0.8 | 1.2 | 0.0 | 0.4 | 0.0 | 4.44 |
| PerAct[18] | 27.9 | 3.71 | 7.2 | 24.0 | 29.2 | 28.8 | 17.71 | 35.6 |
| RVT[14] | 35.4 | 3.28 | 6.4 | 26.0 | 31.3 | 44.8 | 41.1 | 35.3 |
| RVT-2[15] | 56.7 | 1.92 | 15.6 ± 0.8 | 53.0 ± 0.9 | 54.6 ± 0.6 | 59.7 ± 0.7 | 56.7 ± 1.4 | 60.9 ± 0.9 |
| **BridgeVLA (Ours)** | **64.0** | **1.07** | **18.7 ± 2.2** | **60.5 ± 1.1** | **63.8 ± 0.1** | **63.5 ± 1.5** | **68.4 ± 3.3** | **69.3 ± 1.0** |
| **Models** | RO-SIZE | Light Color | Table Color | Table Texture | Distractor | Background Texture | RLBench | Camera Pose |
| R3M-MLP[33] | 0.0 | 1.0 | 1.4 | 0.2 | 1.6 | 1.2 | 2.0 | 0.8 |
| MVP-MLP[46] | 0.0 | 1.6 | 1.6 | 1.0 | 3.8 | 2.2 | 2.0 | 2.6 |
| PerAct[18] | 29.3 | 29.1 | 30.4 | 23.2 | 27.1 | 33.5 | 39.4 | 36.3 |
| RVT[14] | 40.5 | 34.0 | 30.0 | 45.2 | 18.8 | 46.4 | 53.4 | 42.2 |
| RVT-2[15] | 53.4 ± 1.5 | 58.0 ± 1.1 | 62.6 ± 0.9 | 56.6 ± 0.9 | **60.8 ± 0.5** | 68.7 ± 1.1 | 68.8 ± 1.3 | 64.4 ± 0.5 |
| **BridgeVLA (Ours)** | **61.7 ± 0.8** | **69.7 ± 1.2** | **75.7 ± 0.9** | **71.3 ± 0.7** | 51.8 ± 1.5 | **74.8 ± 1.0** | **73.1 ± 0.2** | **73.8 ± 0.3** |

Table 4: **Results on the COLOSSEUM Benchmark.** The table shows the success rates across 14 generalization settings. The "Avg. Rank" column reports the average rank of each method across all perturbations, where lower values indicate better overall performance. Compared to the state-of-the-art baseline, BridgeVLA improves the average success rate by 7.3%.

**Results.** Results are shown in Tab. 4. We use the results of R3M-MLP [33], MVP-MLP [46], RVT [14], and PerAct [39] from the original COLOSSEUM paper [35]. For RVT-2 [15] and BridgeVLA, we perform our own training and evaluation process. We performed three test repetitions and report the average success rate and variance of BridgeVLA and RVT-2 for each task under different perturbations in Tab.6 and Tab.7, respectively. BridgeVLA outperforms all the comparing baseline methods in terms of average success rate, significantly outperforming the best baseline method by 7.3%. Among all the 14 evaluated perturbations, our method ranks the best among all methods in 13 of them. These results address Q3, showcasing that BridgeVLA possesses strong robustness against visual perturbation.

## B.2 Experiments on GemBench

**Setup.** GemBench [12] is a hierarchical generalization benchmark built on the RLBench simulator [19]. Its training set contains 16 tasks (31 variations) covering seven core action primitives—press, pick, push, screw, close, open, and stack/put. The test set consists of 44 tasks (92 variations), categorized into four increasingly challenging settings:

**L1 (Novel Placements)**: L1 consists of the original 16 tasks (31 variations). The object placements are randomized within the workspace. In addition, chromatic distractors are introduced to test the ability to handle additional visual complexity.

**L2 (Novel Rigid Objects)**: L2 involves 15 unseen tasks (28 variations) that require interaction with 8 novel rigid objects using learned primitives. The generalization capabilities are evaluated across two categories: novel object-color compositions and novel object shapes.

**L3 (Novel Articulated Objects)**: L3 consists of 18 unseen tasks (21 variations) that involve interacting with articulated objects. It evaluates the generalization capabilities across three categories: novel action-part compositions, novel object instances, and novel object categories.

**L4 (Novel Long-Horizon Tasks)**: L4 includes 6 complex long-horizon tasks (12 variations) that require combining multiple actions to finish a whole task.

**Baselines.** In total, we compare with six baseline methods. **3D-LOTUS** [12] processes point cloud inputs through a language-conditioned point cloud transformer architecture [45]. It showcases notable multi-tasking capabilities and high training efficiency. Its enhanced variant, **3D-LOTUS++** [12], integrates the generalization capabilities of large-scale models into 3D-LOTUS with a modular architecture consisting of three components: (1) LLM-based task planning [1], (2) VLM-based object grounding [32, 27], and (3) motion control inherited from 3D-LOTUS. We also compare with four methods introduced in Sec. 4.1.1, *i.e.*, **Hiveformer** [16], **PolarNet** [9], **3D Diffuser Actor** [25], **RVT-2** [15]

**Results.** Overall results are shown in Tab. 5 and per-task success rates on the four settings of GemBench are shown in Tab. 8, 9, 10, 11. The results of baseline methods are sourced from [12]. In total, we evaluate on 5 random seeds to reduce statistical variance. And for every seed, we

| Method | Average | L1 | L2 | L3 | L4 |
|---|---|---|---|---|---|
| Hiveformer [16] | 30.4 | $60.3 \pm 1.5$ | $26.1 \pm 1.4$ | $35.1 \pm 1.7$ | $0.0 \pm 0.0$ |
| PolarNet [9] | 38.4 | $77.7 \pm 0.9$ | $37.1 \pm 1.4$ | $38.5 \pm 1.7$ | $0.1 \pm 0.2$ |
| 3D Diffuser Actor [25] | 44.0 | $91.9 \pm 0.8$ | $43.4 \pm 2.8$ | $37.0 \pm 2.2$ | $0.0 \pm 0.0$ |
| RVT-2 [15] | 44.0 | $89.1 \pm 0.8$ | $51.0 \pm 2.3$ | $36.0 \pm 2.2$ | $0.0 \pm 0.0$ |
| 3D-LOTUS [12] | 45.7 | $\mathbf{94.3} \pm 1.4$ | $49.9 \pm 2.2$ | $38.1 \pm 1.1$ | $0.3 \pm 0.3$ |
| 3D-LOTUS++ [12] | 48.0 | $68.7 \pm 0.6$ | $64.5 \pm 0.9$ | $41.5 \pm 1.8$ | $\mathbf{17.4} \pm 0.4$ |
| **BridgeVLA (Ours)** | **50.0** | $91.1 \pm 1.1$ | $\mathbf{65.0} \pm 1.3$ | $\mathbf{43.8} \pm 1.2$ | $0.0 \pm 0.0$ |

Table 5: **Results on GemBench.** We show the average success rates on the four evaluation settings of GemBench. BridgeVLA establishes a new state of the art on this benchmark, achieving an average success rate of 50.0%.

run 20 trials per task variation. BridgeVLA consistently outperforms all the comparing baseline methods in terms of average success rate across the four evaluation settings. Notably, BridgeVLA achieves state-of-the-art results in both the L2 and L3 settings, demonstrating strong generalization capabilities, addressing Q4. However, similar to most baseline approaches, BridgeVLA exhibits limited performance in the L4 setting, where each task comprises multiple sub-tasks. In the future, we plan to explore leveraging large language models (LLMs) for long-horizon task decomposition and further improve the performance in such setting.

### B.3 Key frame Selection

For all the simulation and real-robot experiments, we adopt the same key frame selection strategy as PerAct [39]. A time step is labeled as a key frame if (i) the robot is stationary, (ii) the gripper state changes, or (iii) the step is the final state of the episode. The robot is considered stationary when the absolute velocities of all joints fall below $0.1 \, \mathrm{rad/s}$.

### B.4 Data

Following [39, 14, 15], we select 18 tasks from RLBench [19] to evaluate the performance of our method on complex manipulation tasks. These tasks are visualized in Fig. 5.

To assess the generalization capability of BridgeVLA, we also evaluate on the COLOSSEUM benchmark [35] and GemBench [12]. The COLOSSEUM benchmark includes 20 basic tasks and 12 types of perturbations. These perturbations, which are unseen during training, encompass changes in object texture, color, and size, backgrounds, lighting, distractors and camera poses. The benchmark evaluates on all the 12 types of perturbations, a setting with basic variations from the original RLBench, and a more challenging setting which combines all the 12 types of perturbations. We visualize all perturbations except the one from the original RLBench in Fig. 6.

For GemBench, the training set includes 16 tasks (31 variations) spanning seven fundamental action primitives (press, pick, push, screw, close, open, stack/put). The test set includes 44 tasks (92 variations) organized into four increasingly challenging settings. Unlike RLBench and COLOSSEUM, where demo augmentation is used, we train BridgeVLA using only keyframes from each trajectory without performing any demo augmentation in GemBench.

## C Real-Robot Experiments

### C.1 Experiment Setup

Fig. 3 illustrates our real-robot setting. The platform comprises a 7-DoF Franka Research 3 manipulator with a parallel-jaw gripper and a ZED 2i stereo camera mounted on a tripod for capturing point clouds of the workspace. We collect expert trajectories with a kinestheic teaching approach. We first move the manipulator to keypoints of an expert trajectory and then play back the keypoints to record the observation and action at each keypoint.

## C.2 Basic Setting

This setting provides a scene similar to the training dataset, where only the object layouts are modified. To highlight BridgeVLA's advantages over existing manipulation policies, we compare it with four representative methods in this setting. The behaviors of these baselines are as follows:

**SpatialVLA** [37]: In the experimental setup, we initially trained SpatialVLA using only 10 trajectories per task. However, it failed on nearly all tasks, often struggling to move toward the correct target object. To improve performance, we augmented the dataset with an additional 40 trajectories per task. While this improved performance, it still lagged significantly behind BridgeVLA—particularly on more challenging tasks, such as "Put the giraffe in the lower drawer." These findings suggest that BridgeVLA provides a more effective and data-efficient solution for 3D VLA.

**$\pi_0$** [5]: Similarly, $\pi_0$ fails with only 10 trajectories per task, likely due to overfitting—it performs well on the training set but often fails during online testing. Common failure modes include missing or failing to grasp the target and prematurely opening the gripper before reaching the goal. Notably, both BridgeVLA and $\pi_0$ share the same PaliGemma backbone and are trained end-to-end. This highlights a key contribution of our work: while VLAs like $\pi_0$ perform well with large-scale data, they struggle in low-data regimes—even on simpler tasks, such as "Press sanitizer." In contrast, BridgeVLA achieves near-perfect success and generalizes robustly across diverse settings.

**ACT** [51]: ACT also underperforms compared to BridgeVLA. It demonstrates limited spatial generalization, performing well only in areas densely covered during training, but often failing when the target is near the workspace boundaries. This behavior is consistent with its design: ACT models actions using a Gaussian prior, which assigns low probability to peripheral regions, limiting its spatial generalization capabilities.

**RVT-2** [15]: RVT-2 performs the best among all the baselines. It can successfully solve most tasks, but it is not as robust as BridgeVLA. For instance, it sometimes fails to pick up the block precisely or place the object accurately, leading to task failure. Meanwhile, by utilizing the capabilities of VLM, BridgeVLA's advantages are further amplified in generalization settings, as detailed in Sec. 4.2.

## C.3 Generalization Settings

We evaluate on a total of six generalization settings: Distractor, Lighting, Background, Height, Combination, and Category. For Distractor, Lighting, Background, and Height, we visualize these settings in Fig. 10. We visualize the settings of Combination and Category in Fig. 11 and Fig. 12, respectively.

In Distractor, we add distractor objects that are visually similar to at least one target object to the scene. In Lighting, we evaluate the model in a novel lighting condition in which the lights are off. In Background, we use three different tablecloths to change the background. For Height, we elevate all objects for manipulation with a drawer that is about 10cm high. Distractor, Lighting, Background, and Height aim to evaluate the robustness against visual disturbances.

In Combination, we combine objects and skills that are not paired together in the training datasets. That is, while the object for manipulation and the manipulation skill are seen during training, the instruction that pairs them together is novel. The setting of Combination helps us evaluate whether the model is able to generalize across novel object-skill combinations. In Category, we want to evaluate whether BridgeVLA is able to manipulate objects from categories that are *unseen* in the robot training data. In total, we test 7 novel objects.

## C.4 Preservation of Object Grounding Capability after Fine-tuning

We observe that even after fine-tuning on robot action data, BridgeVLA retains the object grounding capability learned during pre-training. We visualize its predictions on the pre-training dataset after fine-tuning in Fig. 14. It is important to note that the samples in Fig. 14 are not cherry-picked. BridgeVLA does not forget its pre-training knowledge after 3D action fine-tuning.

## C.5 Per-task Success Rate

We showcase per-task success rates of BridgeVLA in the basic setting in Tab. 12. Notably, BridgeVLA achieves exceptionally high success rates even with only 3 trajectories per task, highlighting its superb sample efficiency.

| Task Name | Original | All Perturbations | MO-COLOR | RO-COLOR | MO-TEXTURE | RO-TEXTURE | MO-SIZE | RO-SIZE | Light Color | Table Color | Table Texture | Distractor | Background Texture | RLBench | Camera Pose |
|---|---|---|---|---|---|---|---|---|---|---|---|---|---|---|---|
| basketball_in_hoop | 100.0±0.0 | 4.0±3.3 | 94.7±1.9 | 96.0±0.0 | 84.0±5.7 | - | 100.0±0.0 | 68.0±0.0 | 100.0±0.0 | 100.0±0.0 | 100.0±0.0 | 37.3±1.9 | 100.0±0.0 | 100.0±0.0 | 100.0±0.0 |
| close_box | 100.0±0.0 | 72.0±0.0 | 94.7±1.9 | - | - | - | 93.3±1.9 | - | 100.0±0.0 | 100.0±0.0 | 98.7±1.9 | 98.7±1.9 | 100.0±0.0 | 97.3±1.9 | 100.0±0.0 |
| close_laptop_lid | 100.0±0.0 | 11.1±15.7 | 82.7±3.8 | - | - | - | 67.9±14.6 | - | 89.3±8.2 | 92.0±0.0 | 97.3±3.8 | 82.7±6.8 | 96.0±3.3 | 100.0±0.0 | 96.0±0.0 |
| empty_dishwasher | 0.0±0.0 | 0.0±0.0 | 1.3±1.9 | 1.3±1.9 | - | 1.3±1.9 | 4.0±3.3 | 0.0±0.0 | 0.0±0.0 | 0.0±0.0 | 0.0±0.0 | 0.0±0.0 | 1.3±1.9 | 1.3±1.9 | 0.0±0.0 |
| get_ice_from_fridge | 94.7±1.9 | 5.3±1.9 | 86.7±1.9 | 90.7±7.5 | 90.7±5.0 | - | 84.0±3.3 | 73.3±1.9 | 96.0±3.3 | 98.7±1.9 | 89.3±7.5 | 56.0±8.6 | 94.7±1.9 | 96.0±3.3 | 98.7±1.9 |
| hockey | 57.3±5.0 | 9.3±3.8 | 44.0±6.5 | 50.7±8.2 | - | 50.7±13.2 | 46.7±8.2 | 65.3±5.0 | 45.3±1.9 | 64.0±8.6 | 53.3±1.9 | 20.0±3.3 | 56.0±5.7 | 49.3±5.0 | 50.7±5.0 |
| insert_onto_square_peg | 93.3±3.8 | 23.3±2.4 | 52.0±3.3 | 94.7±1.9 | - | 76.0±8.6 | 85.3±3.8 | 70.7±3.8 | 84.0±0.0 | 88.0±3.3 | 88.0±3.3 | 44.0±11.8 | 86.7±1.9 | 77.3±5.0 | 96.0±0.0 |
| meat_on_grill | 96.0±0.0 | 9.3±1.9 | 32.0±0.0 | 88.0±5.7 | - | - | 100.0±0.0 | - | 100.0±0.0 | 92.0±6.5 | 90.7±1.9 | 98.7±1.9 | 97.3±1.9 | 100.0±0.0 | 100.0±0.0 |
| move_hanger | 37.3±3.8 | 2.7±3.8 | 26.7±3.8 | 46.7±3.8 | - | - | - | - | 52.0±0.0 | 84.0±0.0 | 52.0±5.7 | 52.0±5.7 | 33.3±5.0 | 42.7±1.9 | 24.0±0.0 |
| open_drawer | 96.0±0.0 | 60.0±3.3 | 97.3±1.9 | - | - | - | 90.7±1.9 | - | 88.0±3.3 | 93.3±1.9 | 100.0±0.0 | 90.7±1.9 | 100.0±0.0 | 94.7±1.9 | 96.0±0.0 |
| place_wine_at_rack_location | 88.0±5.7 | 17.3±13.6 | 82.7±5.0 | 89.3±7.5 | - | 92.0±6.5 | 93.3±3.8 | 90.7±3.8 | 90.7±5.0 | 97.3±1.9 | 88.0±3.3 | 74.7±3.8 | 90.7±6.8 | 92.0±3.3 | 92.0±8.6 |
| put_money_in_safe | 94.7±1.9 | 6.7±5.0 | 78.7±1.9 | 74.7±1.9 | 81.3±6.8 | 89.3±5.0 | 92.0±3.3 | - | 37.3±12.4 | 84.0±3.3 | 84.0±3.3 | 84.0±3.3 | 89.3±1.9 | 86.7±8.2 | 86.7±1.9 |
| reach_and_drag | 100.0±0.0 | 0.0±0.0 | 89.3±3.8 | 96.0±0.0 | 94.7±5.0 | 84.0±5.7 | 94.7±1.9 | 38.7±5.0 | 92.0±3.3 | 88.0±5.7 | 78.7±3.8 | 28.0±8.6 | 100.0±0.0 | 100.0±0.0 | 94.7±3.8 |
| scoop_with_spatula | 96.0±3.3 | 6.7±1.9 | 94.7±1.9 | 93.3±1.9 | 85.3±3.8 | 85.3±3.8 | 78.7±3.8 | 86.7±5.0 | 90.7±1.9 | 88.0±6.5 | 77.3±1.9 | 20.0±5.7 | 90.7±6.8 | 89.3±1.9 | 93.3±1.9 |
| setup_chess | 10.7±1.9 | 0.0±0.0 | 1.3±1.9 | 8.0±0.0 | 8.0±3.3 | - | 13.3±1.9 | - | 12.0±5.7 | 21.3±8.2 | 13.3±3.8 | 5.3±1.9 | 20.0±5.7 | 16.0±5.7 | 4.0±3.3 |
| slide_block_to_target | 100.0±0.0 | 24.0±3.3 | 74.7±1.9 | - | 92.0±3.3 | - | 100.0±0.0 | - | 100.0±0.0 | 100.0±0.0 | 98.7±1.9 | 84.0±9.8 | 100.0±0.0 | 100.0±0.0 | 100.0±0.0 |
| stack_cups | 58.7±3.8 | 29.3±1.9 | 66.7±1.9 | - | 50.7±1.9 | - | 44.0±3.3 | - | 62.7±1.9 | 64.0±3.3 | 65.3±8.2 | 26.7±7.5 | 73.3±8.2 | 64.0±14.2 | 72.0±8.6 |
| straighten_rope | 61.3±6.8 | 8.0±5.7 | 16.0±5.7 | - | 48.0±3.3 | - | - | - | 61.3±9.4 | 65.3±1.9 | 54.7±8.2 | 37.3±5.0 | 70.7±8.2 | 66.7±7.5 | 72.0±6.5 |
| turn_oven_on | 93.3±1.9 | 85.3±3.3 | 94.7±3.8 | - | - | - | 90.7±1.9 | - | 93.3±3.8 | 94.7±7.5 | 96.0±3.3 | 96.0±3.3 | 96.0±0.0 | 88.0±3.3 | 100.0±0.0 |
| wipe_desk | 0.0±0.0 | 0.0±0.0 | 0.0±0.0 | 0.0±0.0 | 0.0±0.0 | - | 0.0±0.0 | - | 0.0±0.0 | 0.0±0.0 | 0.0±0.0 | 0.0±0.0 | 0.0±0.0 | 0.0±0.0 | 0.0±0.0 |
| Task Mean | 73.9±0.7 | 18.7±2.2 | 60.5±1.1 | 63.8±0.1 | 63.5±1.5 | 68.4±3.3 | 69.3±1.0 | 61.7±0.0 | 69.7±1.2 | 75.7±0.9 | 71.3±0.7 | 51.8±1.5 | 74.8±1.0 | 73.1±0.2 | 73.8±0.3 |

Table 6: **Success Rates of BridgeVLA under Different Perturbations of COLOSSEUM.**

| Task Name | No variations | All Perturbations | MO-COLOR | RO-COLOR | MO-TEXTURE | RO-TEXTURE | MO-SIZE | RO-SIZE | Light Color | Table Color | Table Texture | Distractor | Background Texture | RLBench | Camera Pose |
|---|---|---|---|---|---|---|---|---|---|---|---|---|---|---|---|
| basketball_in_hoop | 100±0.0 | 40±4.9 | 99±1.7 | 33±1.7 | 96±0.0 | - | 99±1.7 | 100±0.0 | 87±4.4 | 54±2.0 | 91±11.4 | 55±9.1 | 100±0.0 | 97±1.7 | 100±0.0 |
| close_box | 96±4.9 | 32±16.7 | 43±3.3 | - | - | - | 91±5.2 | - | 84±2.8 | 78±3.5 | 91±9.1 | 96±2.8 | 98±3.5 | 96±2.8 | 95±3.3 |
| close_laptop_lid | 30±4.5 | 48±7.5 | 50±4.5 | - | - | - | 56±6.9 | - | 28±2.8 | 23±4.4 | 42±12.8 | 49±5.2 | 44±0.0 | 33±1.7 | 48±2.8 |
| empty_dishwasher | 0±0.0 | 0±0.0 | 0±0.0 | 0±0.0 | - | 1±1.7 | 1±1.7 | 1±1.7 | 0±0.0 | 0±0.0 | 0±0.0 | 0±0.0 | 0±0.0 | 1±1.7 | 1±1.7 |
| get_ice_from_fridge | 66±4.5 | 2±2.0 | 67±3.3 | 11±1.7 | 67±5.2 | - | 71±3.3 | 44±2.8 | 24±0.0 | 35±1.7 | 77±4.4 | 65±3.3 | 70±3.5 | 69±1.7 | 71±4.4 |
| hockey | 12±2.8 | 0±0.0 | 18±6.0 | 0±0.0 | - | 14±3.5 | 2±3.5 | 9±3.3 | 7±5.9 | 10±2.0 | 16±15.0 | 5±1.7 | 13±1.7 | 5±1.7 | 9±4.4 |
| meat_on_grill | 45±1.7 | 56±8.5 | 62±4.5 | 33±1.7 | - | - | 64±2.8 | - | 61±1.7 | 65±1.7 | 49±5.2 | 67±11.1 | 63±3.3 | 62±4.5 | 51±3.3 |
| move_hanger | 0±0.0 | 0±0.0 | 0±0.0 | 0±0.0 | - | - | - | - | 0±0.0 | 0±0.0 | 0±0.0 | 21±9.1 | 0±0.0 | 0±0.0 | 0±0.0 |
| wipe_desk | 0±0.0 | 0±0.0 | 0±0.0 | 0±0.0 | 0±0.0 | - | 0±0.0 | - | 0±0.0 | 0±0.0 | 0±0.0 | 0±0.0 | 0±0.0 | 0±0.0 | 0±0.0 |
| open_drawer | 97±1.7 | 9±5.9 | 100±0.0 | - | - | - | 100±0.0 | - | 90±4.5 | 56±0.0 | 100±0.0 | 89±5.2 | 100±0.0 | 97±1.7 | 96±0.0 |
| slide_block_to_target | 100±0.0 | 37±10.7 | 100±0.0 | - | 100±0.0 | - | 100±0.0 | - | 100±0.0 | 91±1.7 | 88±20.8 | 90±12.8 | 100±0.0 | 100±0.0 | 100±0.0 |
| reach_and_drag | 86±2.0 | 1±1.7 | 34±2.0 | 64±2.8 | 75±4.4 | 74±4.5 | 95±1.7 | 79±1.7 | 20±0.0 | 24±0.0 | 72±15.7 | 43±21.4 | 81±1.7 | 75±1.7 | 80±2.8 |
| put_money_in_safe | 63±1.7 | 1±1.7 | 62±2.0 | 5±1.7 | 58±2.0 | 75±1.7 | 64±2.8 | - | 60±0.0 | 47±3.3 | 82±4.5 | 60±4.9 | 60±0.0 | 60±2.8 | 48±0.0 |
| place_wine_at_rack_location | 96±4.9 | 59±11.4 | 94±4.5 | 94±2.0 | - | 96±2.8 | 91±1.7 | 88±2.8 | 87±5.9 | 93±4.4 | 94±2.0 | 80±16.0 | 88±2.8 | 95±3.3 | 96±4.9 |
| insert_onto_square_peg | 5±1.7 | 0±0.0 | 0±0.0 | 13±1.7 | - | 9±3.3 | 16±0.0 | 6±3.5 | 0±0.0 | 0±0.0 | 0±0.0 | 2±3.5 | 4±0.0 | 4±0.0 | 5±1.7 |
| stack_cups | 44±0.0 | 2±3.5 | 42±2.0 | - | 50±4.5 | - | 8±0.0 | - | 20±0.0 | 13±1.7 | 10±6.0 | 15±3.3 | 40±0.0 | 36±0.0 | 24±0.0 |
| turn_oven_on | 97±1.7 | 5±1.7 | 34±4.5 | - | - | - | 68±2.8 | - | 96±2.8 | 97±1.7 | 98±2.0 | 97±1.7 | 96±2.8 | 92±0.0 | 93±5.2 |
| straighten_rope | 54±2.0 | 0±0.0 | 32±0.0 | - | 57±4.4 | - | - | - | 77±1.7 | 51±4.4 | 14±11.8 | 27±20.3 | 61±1.7 | 66±2.0 | 59±1.7 |
| setup_chess | 5±3.3 | 0±0.0 | 1±1.7 | 4±2.8 | 7±3.3 | - | 4±2.8 | - | 8±4.9 | 4±2.8 | 12±2.8 | 10±10.4 | 18±8.2 | 15±5.2 | 7±4.4 |
| scoop_with_spatula | 96±0.0 | 0±0.0 | 11±1.7 | 73±3.3 | 82±24.2 | 85±1.7 | 81±1.7 | 81±5.2 | 57±1.7 | 87±3.3 | 83±4.4 | 57±11.8 | 93±1.7 | 85±3.3 | 92±2.8 |
| Average | 55±0.5 | 15±1.9 | 42±0.6 | 25±0.2 | 59±2.6 | 51±1.8 | 54±1.0 | 51±1.3 | 45±0.3 | 41±0.0 | 51±2.7 | 46±2.2 | 56±0.5 | 54±0.5 | 54±0.4 |

Table 7: **Success Rates of RVT-2 under Different Perturbations of COLOSSEUM.**

**Table 8: Per-task Success Rate on GemBench Level 1.**

| Method | Avg. | Close Fridge+0 | Close Jar+15 | Close Jar+16 | CloseLaptop Lid+0 | Close Microwave+0 | LightBulb In+17 | LightBulb In+19 | Open Box+0 | Open Door+0 | Open Drawer+0 |
|---|---|---|---|---|---|---|---|---|---|---|---|
| Hiveformer [16] | $60.3_{\pm1.5}$ | $96_{\pm4.2}$ | $64_{\pm13.9}$ | $92_{\pm2.7}$ | $90_{\pm3.5}$ | $88_{\pm7.6}$ | $12_{\pm4.5}$ | $13_{\pm6.7}$ | $4_{\pm4.2}$ | $53_{\pm15.2}$ | $15_{\pm12.2}$ |
| PolarNet [9] | $77.6_{\pm0.9}$ | $99_{\pm2.2}$ | $99_{\pm2.2}$ | $99_{\pm2.2}$ | $95_{\pm3.5}$ | $98_{\pm2.7}$ | $72_{\pm12.5}$ | $71_{\pm6.5}$ | $32_{\pm11.5}$ | $69_{\pm8.9}$ | $61_{\pm12.4}$ |
| 3D diffuser actor [25] | $91.9_{\pm0.8}$ | $\mathbf{100_{\pm0.0}}$ | $\mathbf{100_{\pm0.0}}$ | $\mathbf{100_{\pm0.0}}$ | $99_{\pm2.2}$ | $85_{\pm5.0}$ | $88_{\pm2.7}$ | $98_{\pm2.7}$ | $11_{\pm2.2}$ | $96_{\pm4.2}$ | $82_{\pm9.1}$ |
| RVT-2 [15] | $89.0_{\pm0.8}$ | $77_{\pm11.0}$ | $97_{\pm4.5}$ | $98_{\pm2.7}$ | $77_{\pm13.0}$ | $\mathbf{100_{\pm0.0}}$ | $93_{\pm5.7}$ | $\mathbf{91_{\pm8.2}}$ | $7_{\pm4.5}$ | $\mathbf{98_{\pm4.5}}$ | $\mathbf{93_{\pm5.7}}$ |
| 3D-LOTUS [12] | $\mathbf{94.3_{\pm3.5}}$ | $96_{\pm3.7}$ | $\mathbf{100_{\pm0.0}}$ | $\mathbf{100_{\pm0.0}}$ | $98_{\pm2.5}$ | $98_{\pm4.0}$ | $84_{\pm7.4}$ | $85_{\pm9.5}$ | $\mathbf{99_{\pm2.0}}$ | $77_{\pm2.5}$ | $83_{\pm8.7}$ |
| 3D-LOTUS++ [12] | $68.7_{\pm0.6}$ | $95_{\pm0.0}$ | $\mathbf{100_{\pm0.0}}$ | $99_{\pm2.0}$ | $28_{\pm2.5}$ | $87_{\pm5.1}$ | $55_{\pm10.5}$ | $45_{\pm8.9}$ | $55_{\pm8.9}$ | $79_{\pm9.7}$ | $68_{\pm12.5}$ |
| BridgeVLA (Ours) | $91.1_{\pm1.1}$ | $99_{\pm2.0}$ | $98_{\pm4.0}$ | $\mathbf{100_{\pm0.0}}$ | $97_{\pm2.5}$ | $85_{\pm5.5}$ | $90_{\pm5.5}$ | $87_{\pm7.5}$ | $76_{\pm10.2}$ | $70_{\pm12.3}$ | $86_{\pm5.8}$ |

| Method | Open Drawer+2 | Pick& Lift+0 | Pick& Lift+2 | Pick& Lift+7 | PickUp Cup+8 | PickUp Cup+9 | PickUp Cup+11 | Push Button+0 | Push Button+3 | Push Button+4 | PutIn Cupboard+0 |
|---|---|---|---|---|---|---|---|---|---|---|---|
| Hiveformer [16] | $59_{\pm7.4}$ | $86_{\pm4.2}$ | $92_{\pm6.7}$ | $93_{\pm2.7}$ | $83_{\pm7.6}$ | $69_{\pm12.9}$ | $61_{\pm19.8}$ | $84_{\pm11.9}$ | $68_{\pm6.7}$ | $87_{\pm7.6}$ | $34_{\pm8.2}$ |
| PolarNet [9] | $90_{\pm7.1}$ | $92_{\pm9.1}$ | $84_{\pm7.4}$ | $88_{\pm5.7}$ | $82_{\pm7.6}$ | $79_{\pm4.2}$ | $72_{\pm10.4}$ | $\mathbf{100_{\pm0.0}}$ | $\mathbf{100_{\pm0.0}}$ | $99_{\pm2.2}$ | $52_{\pm7.6}$ |
| 3D diffuser actor [25] | $97_{\pm4.5}$ | $\mathbf{99_{\pm2.2}}$ | $99_{\pm2.2}$ | $99_{\pm2.2}$ | $96_{\pm2.2}$ | $97_{\pm4.5}$ | $98_{\pm2.7}$ | $98_{\pm2.7}$ | $96_{\pm4.2}$ | $98_{\pm2.7}$ | $85_{\pm5.0}$ |
| RVT-2 [15] | $94_{\pm4.2}$ | $\mathbf{99_{\pm2.2}}$ | $98_{\pm2.7}$ | $\mathbf{100_{\pm0.0}}$ | $99_{\pm2.2}$ | $99_{\pm2.2}$ | $99_{\pm2.2}$ | $\mathbf{100_{\pm0.0}}$ | $\mathbf{100_{\pm0.0}}$ | $\mathbf{100_{\pm0.0}}$ | $88_{\pm8.4}$ |
| 3D-LOTUS [12] | $93_{\pm6.0}$ | $\mathbf{99_{\pm2.0}}$ | $\mathbf{100_{\pm0.0}}$ | $99_{\pm2.0}$ | $97_{\pm4.0}$ | $96_{\pm3.7}$ | $94_{\pm4.9}$ | $99_{\pm2.0}$ | $99_{\pm2.0}$ | $\mathbf{100_{\pm0.0}}$ | $\mathbf{89_{\pm5.8}}$ |
| 3D-LOTUS++ [12] | $75_{\pm4.5}$ | $97_{\pm6.0}$ | $94_{\pm3.7}$ | $93_{\pm5.1}$ | $86_{\pm8.0}$ | $88_{\pm6.8}$ | $91_{\pm4.9}$ | $\mathbf{100_{\pm0.0}}$ | $\mathbf{100_{\pm0.0}}$ | $\mathbf{100_{\pm0.0}}$ | $1_{\pm2.0}$ |
| BridgeVLA(Ours) | $99_{\pm2.0}$ | $\mathbf{99_{\pm2.0}}$ | $\mathbf{100_{\pm0.0}}$ | $98_{\pm2.5}$ | $96_{\pm2.0}$ | $94_{\pm3.7}$ | $99_{\pm2.0}$ | $\mathbf{100_{\pm0.0}}$ | $\mathbf{100_{\pm0.0}}$ | $98_{\pm4.0}$ | $74_{\pm6.6}$ |

| Method | PutIn Cupboard+3 | PutMoney InSafe+0 | PutMoney InSafe+1 | Reach& Drag+14 | Reach& Drag+18 | Slide Block+0 | Slide Block+1 | Stack Blocks+30 | Stack Blocks+36 | Stack Blocks+39 |
|---|---|---|---|---|---|---|---|---|---|---|
| Hiveformer [16] | $74_{\pm6.5}$ | $85_{\pm3.5}$ | $88_{\pm2.7}$ | $37_{\pm5.7}$ | $32_{\pm7.6}$ | $99_{\pm2.2}$ | $91_{\pm12.4}$ | $6_{\pm5.5}$ | $7_{\pm4.5}$ | $6_{\pm4.2}$ |
| PolarNet [9] | $\mathbf{88_{\pm4.5}}$ | $93_{\pm4.5}$ | $95_{\pm5.0}$ | $99_{\pm2.2}$ | $99_{\pm2.2}$ | $\mathbf{100_{\pm0.0}}$ | $0_{\pm0.0}$ | $34_{\pm10.8}$ | $30_{\pm9.4}$ | $36_{\pm12.9}$ |
| 3D diffuser actor [25] | $82_{\pm11.5}$ | $\mathbf{95_{\pm5.0}}$ | $98_{\pm2.7}$ | $\mathbf{100_{\pm0.0}}$ | $99_{\pm2.2}$ | $\mathbf{100_{\pm0.0}}$ | $89_{\pm4.2}$ | $88_{\pm7.6}$ | $85_{\pm6.1}$ | $89_{\pm5.5}$ |
| RVT-2 [15] | $80_{\pm6.1}$ | $93_{\pm4.4}$ | $96_{\pm5.8}$ | $85_{\pm10.0}$ | $94_{\pm2.2}$ | $\mathbf{100_{\pm0.0}}$ | $\mathbf{100_{\pm0.0}}$ | $88_{\pm5.7}$ | $\mathbf{93_{\pm2.7}}$ | $88_{\pm11.5}$ |
| 3D-LOTUS [12] | $72_{\pm11.2}$ | $94_{\pm3.7}$ | $\mathbf{99_{\pm2.0}}$ | $99_{\pm2.0}$ | $\mathbf{100_{\pm0.0}}$ | $\mathbf{100_{\pm0.0}}$ | $\mathbf{100_{\pm0.0}}$ | $\mathbf{94_{\pm5.8}}$ | $91_{\pm6.6}$ | $\mathbf{90_{\pm4.5}}$ |
| 3D-LOTUS++ [12] | $2_{\pm2.5}$ | $22_{\pm6.8}$ | $16_{\pm4.9}$ | $94_{\pm3.7}$ | $62_{\pm8.7}$ | $\mathbf{100_{\pm0.0}}$ | $65_{\pm5.5}$ | $86_{\pm5.8}$ | $20_{\pm4.5}$ | $28_{\pm13.6}$ |
| BridgeVLA (Ours) | $84_{\pm6.6}$ | $79_{\pm9.7}$ | $86_{\pm3.7}$ | $96_{\pm5.8}$ | $97_{\pm4.0}$ | $\mathbf{100_{\pm0.0}}$ | $90_{\pm5.5}$ | $77_{\pm8.1}$ | $87_{\pm4.0}$ | $85_{\pm7.8}$ |

**Table 9: Per-task Success Rate on GemBench Level 2.**

| Method | Avg. | Push Button+13 | Push Button+15 | Push Button+17 | Pick& Lift+14 | Pick& Lift+16 | Pick& Lift+18 | PickUp Cup+10 | PickUp Cup+12 | PickUp Cup+13 |
|---|---|---|---|---|---|---|---|---|---|---|
| Hiveformer | $26.1_{\pm1.4}$ | $97_{\pm2.7}$ | $85_{\pm10.0}$ | $88_{\pm2.7}$ | $21_{\pm6.5}$ | $9_{\pm4.2}$ | $8_{\pm6.7}$ | $30_{\pm7.1}$ | $22_{\pm13.5}$ | $26_{\pm10.6}$ |
| PolarNet | $37.1_{\pm1.4}$ | $\mathbf{100_{\pm0.0}}$ | $\mathbf{100_{\pm0.0}}$ | $85_{\pm7.9}$ | $3_{\pm4.5}$ | $1_{\pm2.2}$ | $0_{\pm0.0}$ | $48_{\pm5.5}$ | $46_{\pm8.9}$ | $16_{\pm6.5}$ |
| 3D diffuser actor | $43.4_{\pm2.8}$ | $87_{\pm13.0}$ | $81_{\pm6.5}$ | $60_{\pm9.4}$ | $9_{\pm4.2}$ | $18_{\pm9.1}$ | $0_{\pm0.0}$ | $84_{\pm5.5}$ | $60_{\pm11.7}$ | $62_{\pm13.0}$ |
| RVT-2 | $51.0_{\pm2.3}$ | $\mathbf{100_{\pm0.0}}$ | $\mathbf{100_{\pm0.0}}$ | $\mathbf{100_{\pm0.0}}$ | $47_{\pm7.6}$ | $29_{\pm9.6}$ | $8_{\pm4.5}$ | $81_{\pm8.2}$ | $59_{\pm9.6}$ | $72_{\pm9.7}$ |
| 3D-LOTUS | $49.9_{\pm2.2}$ | $99_{\pm2.0}$ | $\mathbf{100_{\pm0.0}}$ | $\mathbf{100_{\pm0.0}}$ | $3_{\pm2.5}$ | $18_{\pm8.7}$ | $33_{\pm9.3}$ | $89_{\pm3.7}$ | $78_{\pm8.7}$ | $57_{\pm7.5}$ |
| 3D-LOTUS++ | $64.5_{\pm0.9}$ | $99_{\pm2.0}$ | $\mathbf{100_{\pm0.0}}$ | $99_{\pm2.0}$ | $\mathbf{94_{\pm3.7}}$ | $\mathbf{96_{\pm3.7}}$ | $\mathbf{95_{\pm3.2}}$ | $79_{\pm4.9}$ | $89_{\pm9.7}$ | $84_{\pm10.2}$ |
| BridgeVLA (Ours) | $\mathbf{65.0_{\pm1.3}}$ | $\mathbf{100_{\pm0.0}}$ | $\mathbf{100_{\pm0.0}}$ | $\mathbf{100_{\pm0.0}}$ | $74_{\pm9.7}$ | $89_{\pm4.9}$ | $0_{\pm0.0}$ | $\mathbf{91_{\pm3.7}}$ | $\mathbf{90_{\pm3.2}}$ | $\mathbf{90_{\pm6.3}}$ |

| Method | Stack Blocks+24 | Stack Blocks+27 | Stack Blocks+33 | Slide Block+2 | Slide Block+3 | Close Jar+3 | Close Jar+4 | LightBulb In+1 | LightBulb In+2 | Lamp On+0 |
|---|---|---|---|---|---|---|---|---|---|---|
| Hiveformer | $0_{\pm0.0}$ | $4_{\pm4.2}$ | $0_{\pm0.0}$ | $0_{\pm0.0}$ | $0_{\pm0.0}$ | $0_{\pm0.0}$ | $0_{\pm0.0}$ | $4_{\pm4.2}$ | $0_{\pm0.0}$ | $7_{\pm4.5}$ |
| PolarNet | $1_{\pm2.2}$ | $2_{\pm2.7}$ | $6_{\pm8.2}$ | $0_{\pm0.0}$ | $0_{\pm0.0}$ | $20_{\pm10.6}$ | $82_{\pm5.7}$ | $22_{\pm11.5}$ | $17_{\pm8.4}$ | $\mathbf{14_{\pm10.8}}$ |
| 3D diffuser actor | $\mathbf{66_{\pm13.9}}$ | $82_{\pm2.7}$ | $50_{\pm16.6}$ | $0_{\pm0.0}$ | $0_{\pm0.0}$ | $23_{\pm16.4}$ | $37_{\pm6.7}$ | $51_{\pm17.8}$ | $60_{\pm10.0}$ | $7_{\pm7.6}$ |
| RVT-2 | $18_{\pm4.5}$ | $56_{\pm16.7}$ | $45_{\pm13.7}$ | $0_{\pm0.0}$ | $1_{\pm2.2}$ | $7_{\pm7.6}$ | $77_{\pm5.7}$ | $\mathbf{68_{\pm14.4}}$ | $6_{\pm6.5}$ | $0_{\pm0.0}$ |
| 3D-LOTUS | $13_{\pm8.1}$ | $40_{\pm9.5}$ | $69_{\pm5.8}$ | $0_{\pm0.0}$ | $0_{\pm0.0}$ | $71_{\pm5.8}$ | $90_{\pm4.5}$ | $24_{\pm4.9}$ | $41_{\pm8.6}$ | $0_{\pm0.0}$ |
| 3D-LOTUS++ | $22_{\pm9.3}$ | $\mathbf{83_{\pm7.5}}$ | $59_{\pm3.7}$ | $\mathbf{27_{\pm9.8}}$ | $\mathbf{5_{\pm3.2}}$ | $\mathbf{98_{\pm2.5}}$ | $\mathbf{96_{\pm3.7}}$ | $56_{\pm9.7}$ | $43_{\pm7.5}$ | $2_{\pm2.0}$ |
| BridgeVLA (Ours) | $61_{\pm10.7}$ | $51_{\pm13.2}$ | $\mathbf{79_{\pm8.6}}$ | $12_{\pm9.3}$ | $3_{\pm4.0}$ | $66_{\pm6.6}$ | $88_{\pm4.0}$ | $66_{\pm8.6}$ | $\mathbf{74_{\pm5.8}}$ | $7_{\pm4.0}$ |

| Method | Reach& Drag+5 | Reach& Drag+7 | PutCube InSafe+0 | Pick&Lift Cylinder+0 | Pick&Lift Star+0 | Pick&Lift Moon+0 | Pick&Lift Toy+0 | PutIn Cupboard+7 | PutIn Cupboard+8 |
|---|---|---|---|---|---|---|---|---|---|
| Hiveformer | $1_{\pm2.2}$ | $0_{\pm0.0}$ | $4_{\pm2.2}$ | $78_{\pm5.7}$ | $73_{\pm7.6}$ | $88_{\pm2.7}$ | $87_{\pm4.5}$ | $0_{\pm0.0}$ | $0_{\pm0.0}$ |
| PolarNet | $61_{\pm8.2}$ | $10_{\pm6.1}$ | $\mathbf{40_{\pm14.1}}$ | $93_{\pm6.7}$ | $88_{\pm8.4}$ | $93_{\pm6.7}$ | $90_{\pm3.5}$ | $0_{\pm0.0}$ | $0_{\pm0.0}$ |
| 3D diffuser actor | $0_{\pm0.0}$ | $64_{\pm6.5}$ | $3_{\pm2.7}$ | $\mathbf{99_{\pm2.2}}$ | $43_{\pm17.9}$ | $91_{\pm9.6}$ | $30_{\pm9.4}$ | $0_{\pm0.0}$ | $\mathbf{3_{\pm4.5}}$ |
| RVT-2 | $91_{\pm2.2}$ | $89_{\pm6.5}$ | $6_{\pm5.5}$ | $98_{\pm2.7}$ | $98_{\pm4.5}$ | $94_{\pm4.2}$ | $78_{\pm8.4}$ | $0_{\pm0.0}$ | $0_{\pm0.0}$ |
| 3D-LOTUS | $\mathbf{95_{\pm4.5}}$ | $18_{\pm10.8}$ | $25_{\pm5.5}$ | $88_{\pm8.7}$ | $69_{\pm6.6}$ | $80_{\pm8.4}$ | $\mathbf{96_{\pm3.7}}$ | $0_{\pm0.0}$ | $0_{\pm0.0}$ |
| 3D-LOTUS++ | $94_{\pm2.0}$ | $64_{\pm12.4}$ | $37_{\pm5.1}$ | $91_{\pm2.0}$ | $94_{\pm3.7}$ | $29_{\pm6.6}$ | $71_{\pm2.0}$ | $\mathbf{1_{\pm2.0}}$ | $0_{\pm0.0}$ |
| BridgeVLA (Ours) | $94_{\pm3.7}$ | $\mathbf{96_{\pm3.7}}$ | $3_{\pm2.5}$ | $98_{\pm2.5}$ | $\mathbf{99_{\pm2.0}}$ | $95_{\pm3.2}$ | $93_{\pm5.1}$ | $0_{\pm0.0}$ | $0_{\pm0.0}$ |

**Table 10: Per-task Success Rate on GemBench Level 3.**

| Method | Avg. | Close Door+0 | Close Box+0 | Close Fridge2+0 | CloseLaptop Lid2+0 | Close Microwave2+0 | Open Door2+0 | Open Box2+0 |
|---|---|---|---|---|---|---|---|---|
| Hiveformer | $35.1_{\pm1.7}$ | $0_{\pm0.0}$ | $1_{\pm2.2}$ | $34_{\pm9.6}$ | $52_{\pm9.1}$ | $15_{\pm7.1}$ | $32_{\pm11.5}$ | $5_{\pm3.5}$ |
| PolarNet | $38.5_{\pm1.7}$ | $0_{\pm0.0}$ | $0_{\pm0.0}$ | $78_{\pm5.7}$ | $26_{\pm8.2}$ | $74_{\pm6.5}$ | $33_{\pm6.7}$ | $23_{\pm8.4}$ |
| 3D diffuser actor | $37.0_{\pm2.2}$ | $0_{\pm0.0}$ | $0_{\pm0.0}$ | $97_{\pm2.7}$ | $23_{\pm6.7}$ | $88_{\pm7.6}$ | $\mathbf{86}_{\pm7.4}$ | $67_{\pm9.8}$ |
| RVT-2 | $36.0_{\pm2.2}$ | $1_{\pm2.2}$ | $2_{\pm2.7}$ | $72_{\pm6.7}$ | $42_{\pm14.0}$ | $71_{\pm8.9}$ | $79_{\pm6.5}$ | $5_{\pm6.1}$ |
| 3D-LOTUS | $38.1_{\pm1.1}$ | $0_{\pm0.0}$ | $\mathbf{58}_{\pm8.1}$ | $36_{\pm9.7}$ | $54_{\pm10.7}$ | $85_{\pm7.1}$ | $42_{\pm6.8}$ | $11_{\pm6.6}$ |
| 3D-LOTUS++ | $41.5_{\pm1.8}$ | $\mathbf{1}_{\pm2.0}$ | $29_{\pm8.6}$ | $93_{\pm2.5}$ | $50_{\pm9.5}$ | $\mathbf{99}_{\pm2.0}$ | $52_{\pm10.3}$ | $16_{\pm8.0}$ |
| BridgeVLA (Ours) | $\mathbf{43.8}_{\pm1.2}$ | $0_{\pm0.0}$ | $1_{\pm2.0}$ | $95_{\pm5.5}$ | $\mathbf{77}_{\pm4.0}$ | $54_{\pm10.2}$ | $68_{\pm10.8}$ | $\mathbf{74}_{\pm4.9}$ |

| Method | Open Drawer2+0 | Open Drawer3+0 | OpenDrawer Long+0 | OpenDrawer Long+1 | OpenDrawer Long+2 | OpenDrawer Long+3 | Toilet SeatUp+0 | Open Fridge+0 |
|---|---|---|---|---|---|---|---|---|
| Hiveformer | $59_{\pm11.9}$ | $39_{\pm11.9}$ | $78_{\pm8.4}$ | $82_{\pm4.5}$ | $49_{\pm4.2}$ | $57_{\pm11.5}$ | $6_{\pm4.2}$ | $0_{\pm0.0}$ |
| PolarNet | $\mathbf{91}_{\pm4.2}$ | $29_{\pm8.2}$ | $84_{\pm11.9}$ | $\mathbf{88}_{\pm5.7}$ | $\mathbf{63}_{\pm8.4}$ | $37_{\pm7.6}$ | $2_{\pm2.7}$ | $4_{\pm2.2}$ |
| 3D diffuser actor | $19_{\pm8.2}$ | $1_{\pm2.2}$ | $15_{\pm5.0}$ | $35_{\pm13.7}$ | $26_{\pm9.6}$ | $79_{\pm12.9}$ | $0_{\pm0.0}$ | $7_{\pm5.7}$ |
| RVT-2 | $81_{\pm11.9}$ | $0_{\pm0.0}$ | $\mathbf{84}_{\pm8.2}$ | $39_{\pm10.8}$ | $11_{\pm8.9}$ | $75_{\pm6.1}$ | $7_{\pm5.7}$ | $0_{\pm0.0}$ |
| 3D-LOTUS | $90_{\pm3.2}$ | $22_{\pm8.1}$ | $56_{\pm13.9}$ | $33_{\pm11.2}$ | $17_{\pm8.1}$ | $75_{\pm6.3}$ | $0_{\pm0.0}$ | $4_{\pm5.8}$ |
| 3D-LOTUS++ | $70_{\pm5.5}$ | $41_{\pm4.9}$ | $72_{\pm4.0}$ | $52_{\pm10.8}$ | $23_{\pm8.1}$ | $78_{\pm5.1}$ | $\mathbf{8}_{\pm5.1}$ | $0_{\pm0.0}$ |
| BridgeVLA (Ours) | $65_{\pm6.3}$ | $\mathbf{87}_{\pm6.0}$ | $59_{\pm8.6}$ | $34_{\pm8.0}$ | $18_{\pm10.3}$ | $\mathbf{85}_{\pm8.4}$ | $6_{\pm5.8}$ | $7_{\pm2.5}$ |

| Method | OpenLaptop Lid+0 | Open Microwave+0 | PutMoney InSafe+2 | Open Drawer+1 | Close Drawer+0 | Close Grill+0 |
|---|---|---|---|---|---|---|
| Hiveformer | $\mathbf{100}_{\pm0.0}$ | $0_{\pm0.0}$ | $0_{\pm0.0}$ | $0_{\pm0.0}$ | $83_{\pm5.7}$ | $44_{\pm10.8}$ |
| PolarNet | $\mathbf{100}_{\pm0.0}$ | $0_{\pm0.0}$ | $1_{\pm2.2}$ | $4_{\pm4.2}$ | $29_{\pm11.9}$ | $42_{\pm11.5}$ |
| 3D diffuser actor | $\mathbf{100}_{\pm0.0}$ | $0_{\pm0.0}$ | $2_{\pm4.5}$ | $0_{\pm0.0}$ | $66_{\pm7.4}$ | $\mathbf{65}_{\pm13.7}$ |
| RVT-2 | $93_{\pm5.7}$ | $0_{\pm0.0}$ | $0_{\pm0.0}$ | $\mathbf{6}_{\pm2.2}$ | $78_{\pm8.4}$ | $9_{\pm4.2}$ |
| 3D-LOTUS | $\mathbf{100}_{\pm0.0}$ | $0_{\pm0.0}$ | $0_{\pm0.0}$ | $0_{\pm0.0}$ | $\mathbf{87}_{\pm8.1}$ | $29_{\pm6.6}$ |
| 3D-LOTUS++ | $86_{\pm6.6}$ | $0_{\pm0.0}$ | $\mathbf{13}_{\pm8.1}$ | $0_{\pm0.0}$ | $69_{\pm5.8}$ | $19_{\pm13.9}$ |
| BridgeVLA (Ours) | $95_{\pm0.0}$ | $0_{\pm0.0}$ | $2_{\pm2.5}$ | $0_{\pm0.0}$ | $58_{\pm12.9}$ | $35_{\pm12.3}$ |

Table 10: Per-task Success Rate on GemBench Level 3.

**Table 11: Per-task Success Rate on GemBench Level 4.**

| Method | Avg. | Push Buttons4+1 | Push Buttons4+2 | Push Buttons4+3 | TakeShoes OutOfBox+0 | PutItems InDrawer+0 | PutItems InDrawer+2 |
|---|---|---|---|---|---|---|---|
| Hiveformer | $0_{\pm0.0}$ | $0_{\pm0.0}$ | $0_{\pm0.0}$ | $0_{\pm0.0}$ | $0_{\pm0.0}$ | $0_{\pm0.0}$ | $0_{\pm0.0}$ |
| PolarNet | $0.1_{\pm0.2}$ | $1_{\pm2.2}$ | $0_{\pm0.0}$ | $0_{\pm0.0}$ | $0_{\pm0.0}$ | $0_{\pm0.0}$ | $0_{\pm0.0}$ |
| 3D diffuser actor | $0_{\pm0.0}$ | $0_{\pm0.0}$ | $0_{\pm0.0}$ | $0_{\pm0.0}$ | $0_{\pm0.0}$ | $0_{\pm0.0}$ | $0_{\pm0.0}$ |
| RVT-2 | $0_{\pm0.0}$ | $0_{\pm0.0}$ | $0_{\pm0.0}$ | $0_{\pm0.0}$ | $0_{\pm0.0}$ | $0_{\pm0.0}$ | $0_{\pm0.0}$ |
| 3D-LOTUS | $0.3_{\pm0.3}$ | $3_{\pm4.0}$ | $0_{\pm0.0}$ | $0_{\pm0.0}$ | $0_{\pm0.0}$ | $0_{\pm0.0}$ | $0_{\pm0.0}$ |
| 3D-LOTUS++ | $\mathbf{17.4}_{\pm0.4}$ | $\mathbf{76}_{\pm7.4}$ | $\mathbf{49}_{\pm8.6}$ | $\mathbf{37}_{\pm8.1}$ | $0_{\pm0.0}$ | $0_{\pm0.0}$ | $0_{\pm0.0}$ |
| BridgeVLA (Ours) | $0_{\pm0.0}$ | $0_{\pm0.0}$ | $0_{\pm0.0}$ | $0_{\pm0.0}$ | $0_{\pm0.0}$ | $0_{\pm0.0}$ | $0_{\pm0.0}$ |

| Method | PutItems InDrawer+4 | Tower4+1 | Tower4+3 | Stack Cups+0 | Stack Cups+3 | PutAllGroceries InCupboard+0 |
|---|---|---|---|---|---|---|
| Hiveformer | $0_{\pm0.0}$ | $0_{\pm0.0}$ | $0_{\pm0.0}$ | $0_{\pm0.0}$ | $0_{\pm0.0}$ | $0_{\pm0.0}$ |
| PolarNet | $0_{\pm0.0}$ | $0_{\pm0.0}$ | $0_{\pm0.0}$ | $0_{\pm0.0}$ | $0_{\pm0.0}$ | $0_{\pm0.0}$ |
| 3D diffuser actor | $0_{\pm0.0}$ | $0_{\pm0.0}$ | $0_{\pm0.0}$ | $0_{\pm0.0}$ | $0_{\pm0.0}$ | $0_{\pm0.0}$ |
| RVT-2 | $0_{\pm0.0}$ | $0_{\pm0.0}$ | $0_{\pm0.0}$ | $0_{\pm0.0}$ | $0_{\pm0.0}$ | $0_{\pm0.0}$ |
| 3D-LOTUS | $0_{\pm0.0}$ | $0_{\pm0.0}$ | $0_{\pm0.0}$ | $0_{\pm0.0}$ | $0_{\pm0.0}$ | $0_{\pm0.0}$ |
| 3D-LOTUS++ | $0_{\pm0.0}$ | $\mathbf{17}_{\pm10.8}$ | $\mathbf{30}_{\pm13.4}$ | $0_{\pm0.0}$ | $0_{\pm0.0}$ | $0_{\pm0.0}$ |
| BridgeVLA (Ours) | $0_{\pm0.0}$ | $0_{\pm0.0}$ | $0_{\pm0.0}$ | $0_{\pm0.0}$ | $0_{\pm0.0}$ | $0_{\pm0.0}$ |

Table 11: Per-task Success Rate on GemBench Level 4.

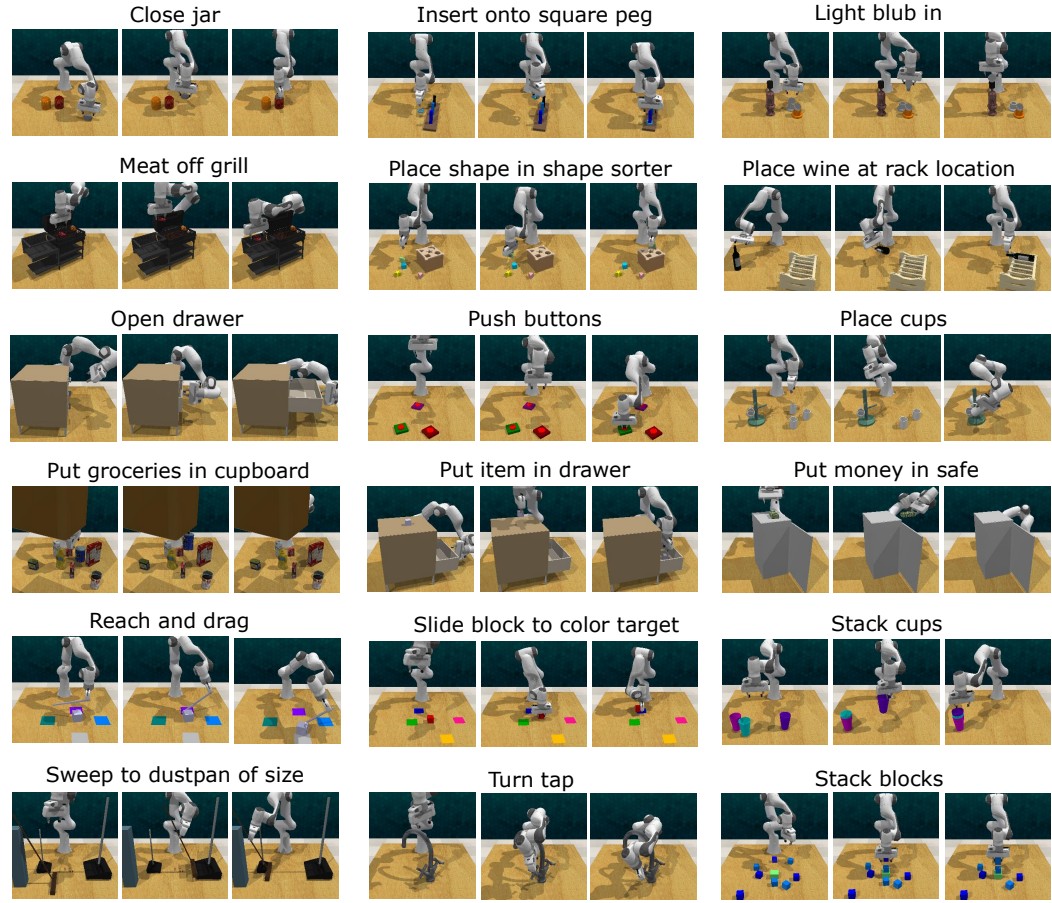

Figure 5: **Visualization of 18 RLBench [19] Tasks**.

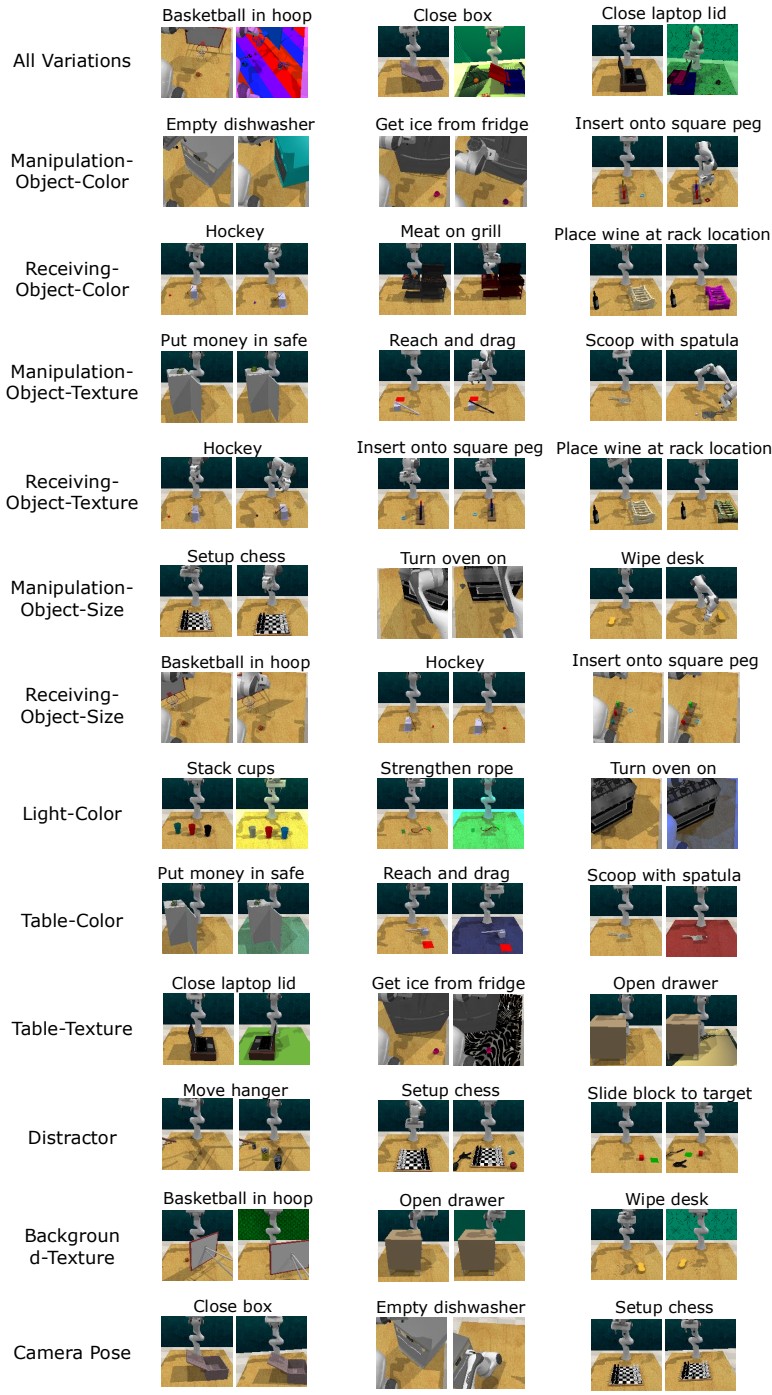

Figure 6: **Visualization of Perturbations in COLOSSEUM** [35].

| Task | 3 trajectories | 10 trajectories |
|------|----------------|-----------------|
| Put the RedBull can in the top shelf | 9/10 | 10/10 |
| Put the soda can in the bottom shelf | 9/10 | 9/10 |
| Put the RedBull can in the bottom shelf | 10/10 | 10/10 |
| Put the coke can in the top shelf | 10/10 | 10/10 |
| Place the red block in the blue plate | 10/10 | 10/10 |
| Place the orange block in the green plate | 10/10 | 10/10 |
| Put the wolf in the upper drawer | 7/10 | 9/10 |
| Place the red block in the purple plate | 10/10 | 10/10 |
| Place the yellow block in the green plate | 10/10 | 10/10 |
| Press sanitizer | 10/10 | 10/10 |
| Put the zebra in the upper drawer | 9/10 | 9/10 |
| Put the giraffe in the lower drawer | 10/10 | 9/10 |
| Put the zebra in the lower drawer | 10/10 | 10/10 |

Table 12: **Per-task Success Rates of BridgeVLA in the Basic Setting.**

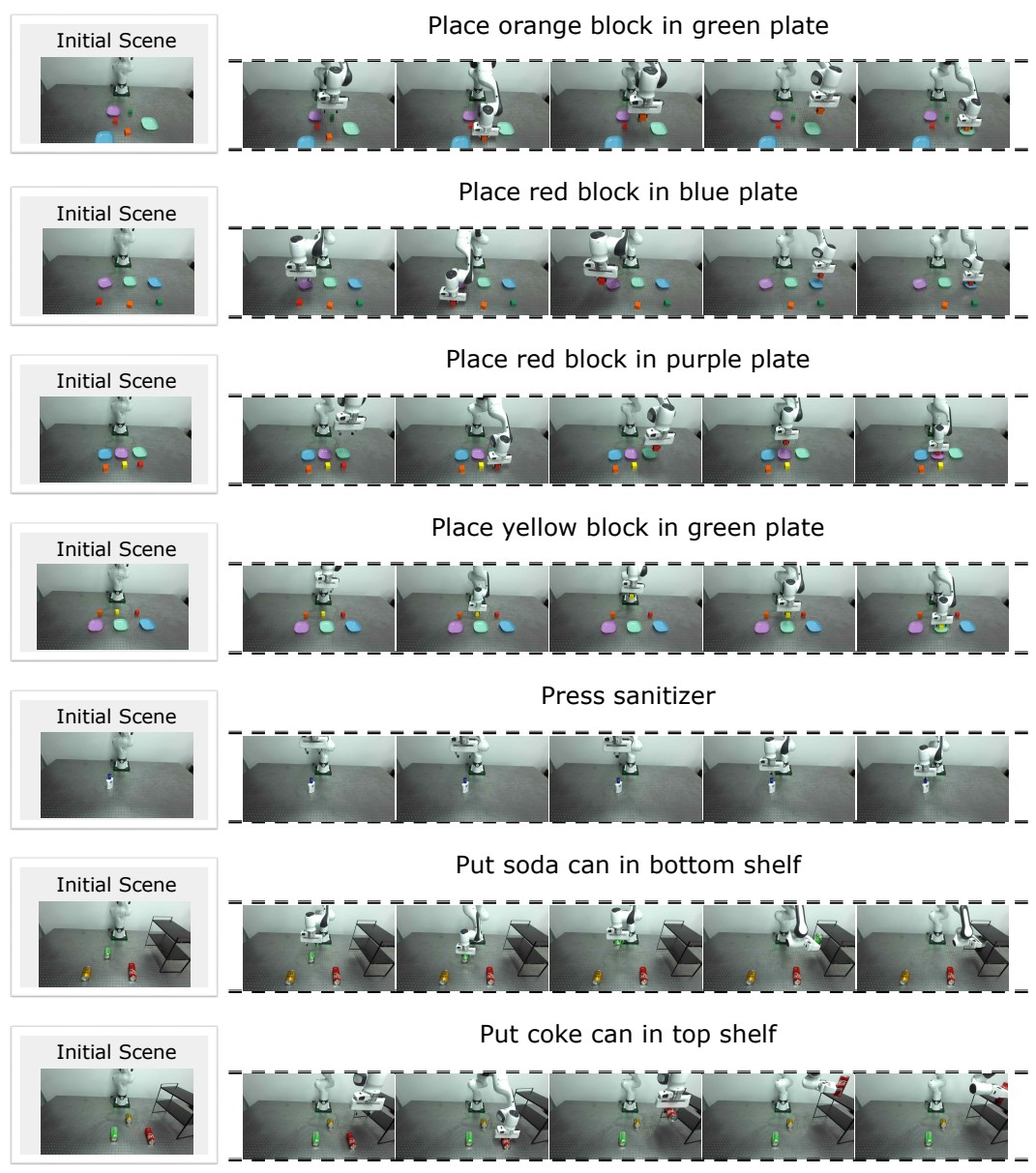

Figure 7: **Real-Robot Rollouts (I).**

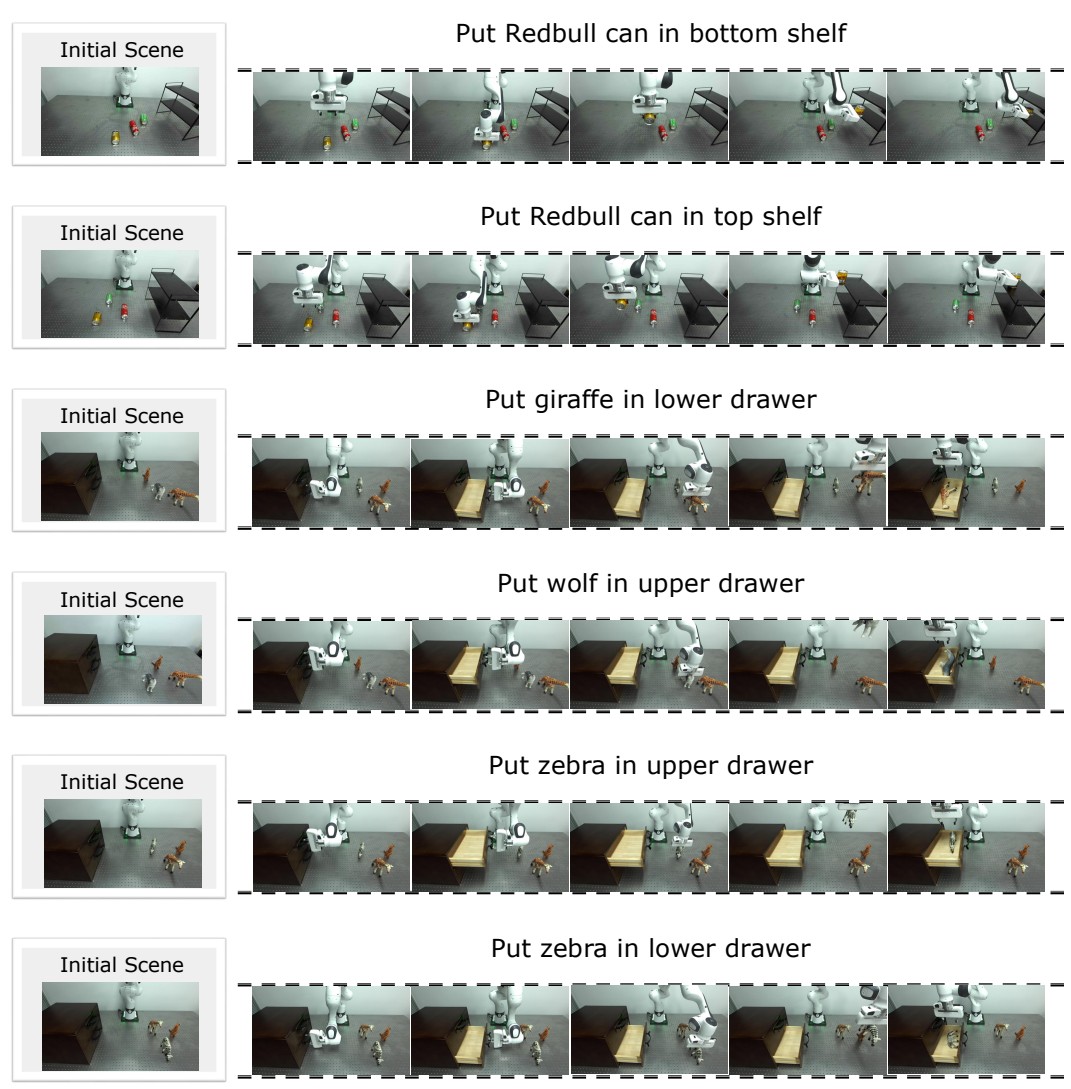

Figure 8: **Real-Robot Rollouts (II).**

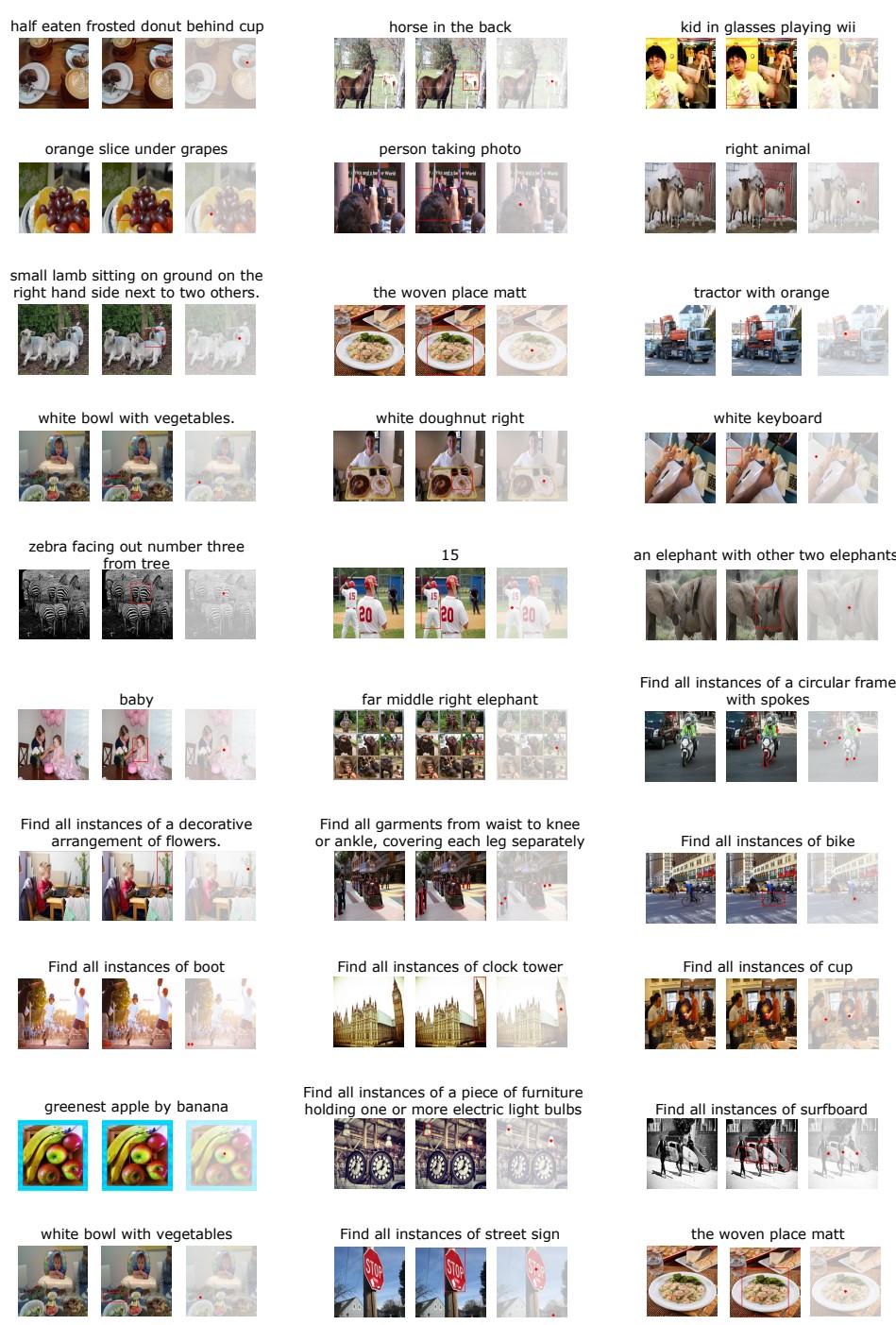

Figure 9: **Visualization of Pre-training Data.** We list some samples of pre-training data. For every sample, the left shows the original image; the middle shows the bounding boxes of the objects of interest; the right shows the ground-truth heatmap used for training.

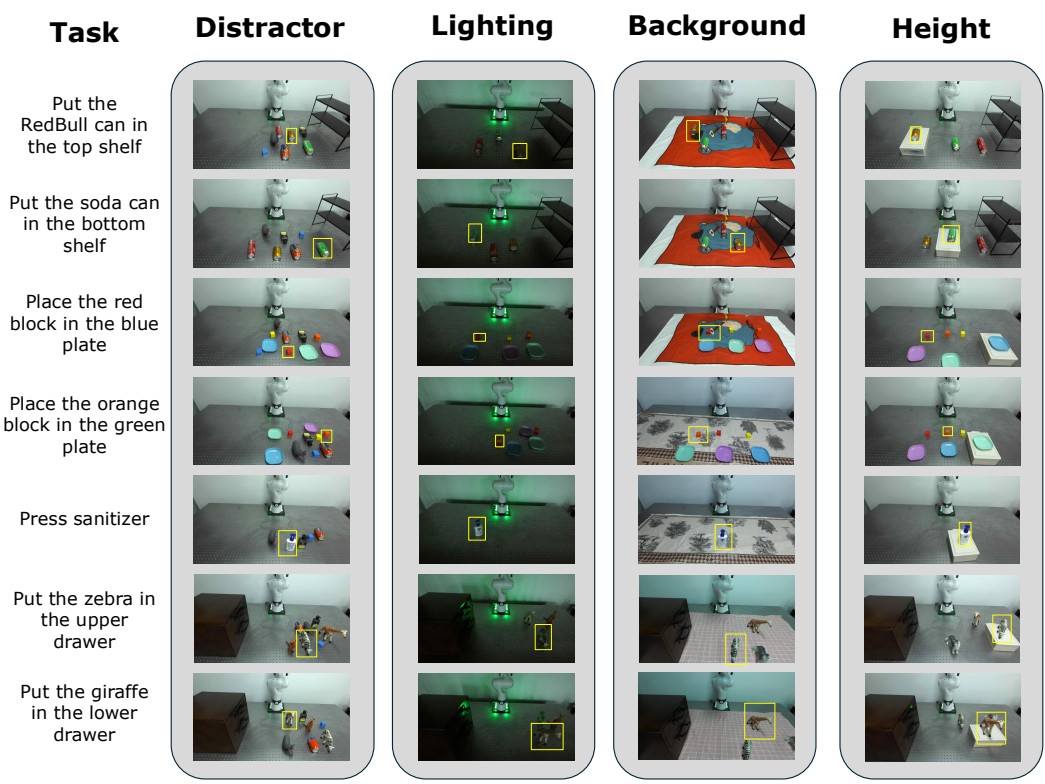

Figure 10: **Visualization of the Distractor, Lighting, Background, and Height settings.**

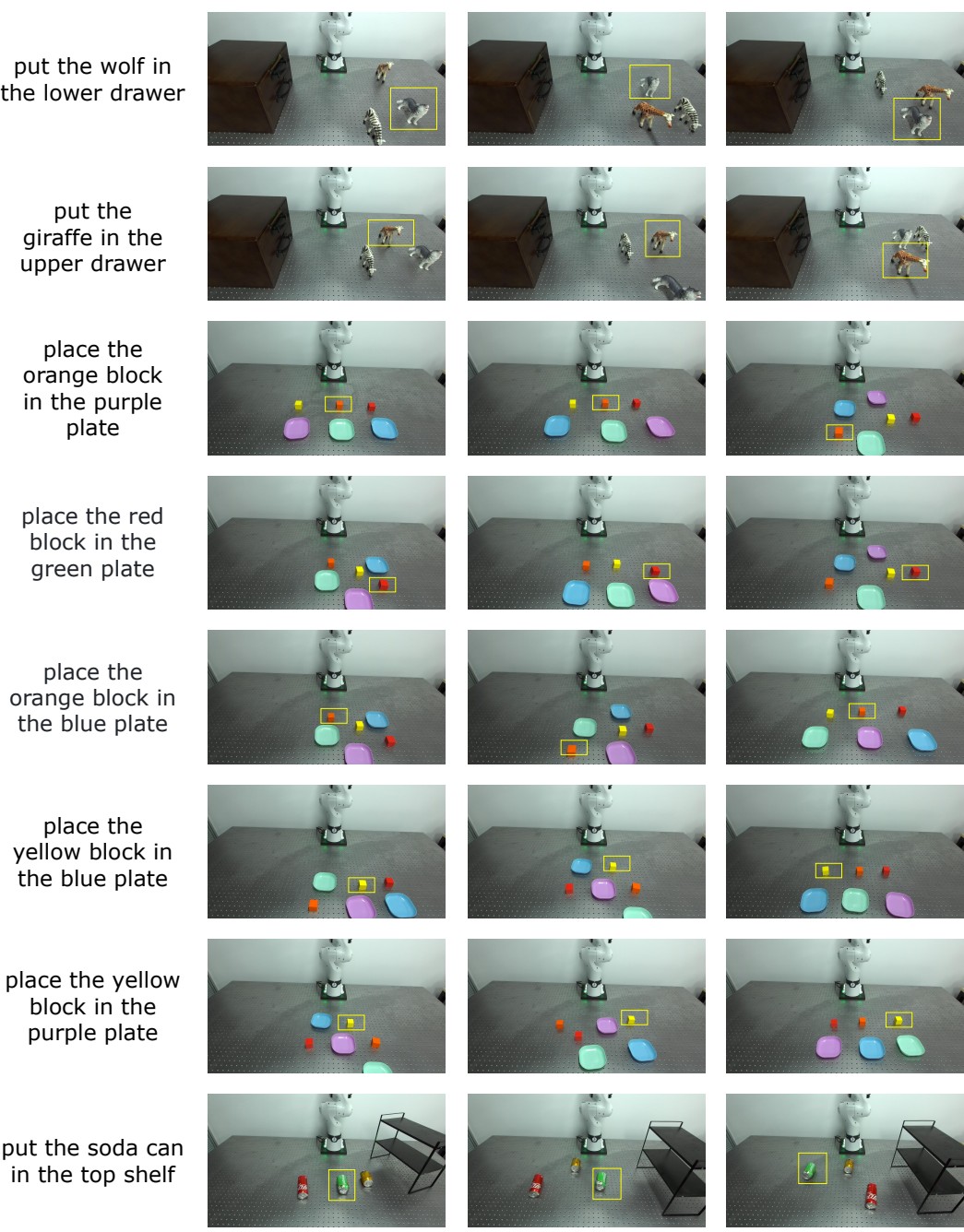

Figure 11: **Visualization of the Combination Setting (I).** During training, the manipulated objects and skills are seen, but their combinations are unseen.

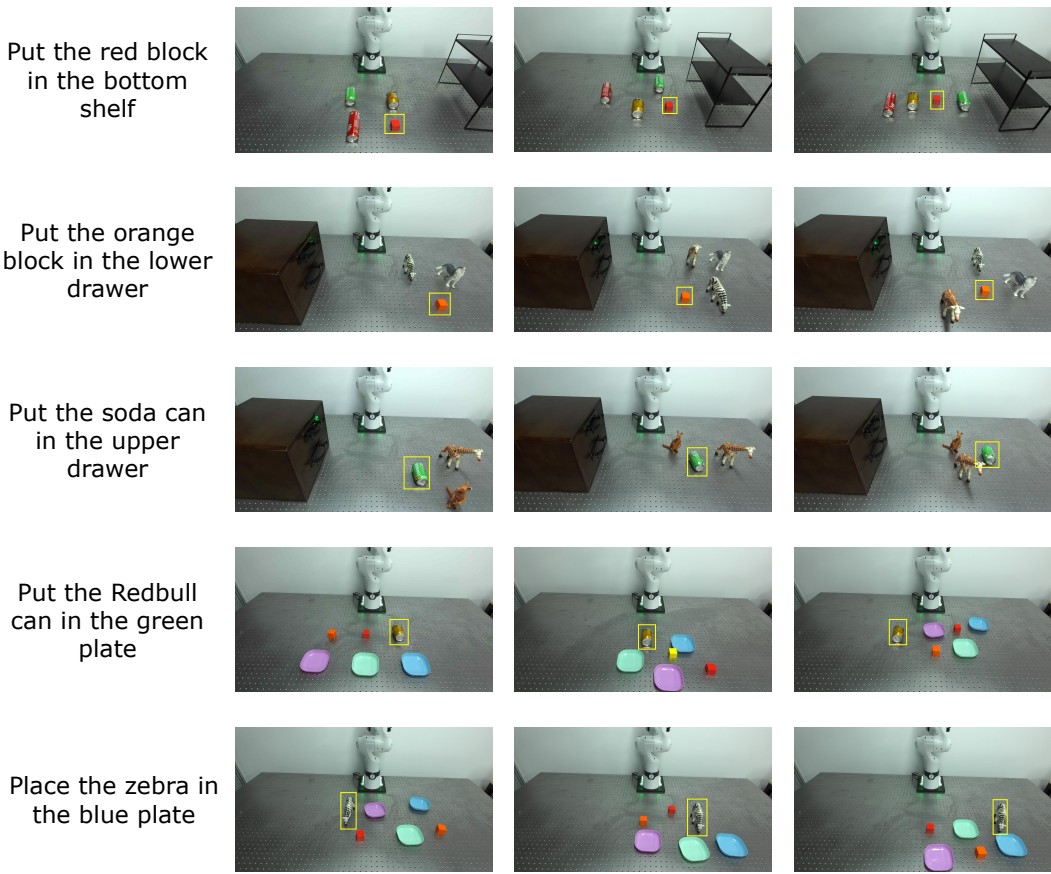

Figure 12: **Visualization of the Combination Setting (II).** During training, the manipulated objects and skills are seen, but their combinations are unseen.

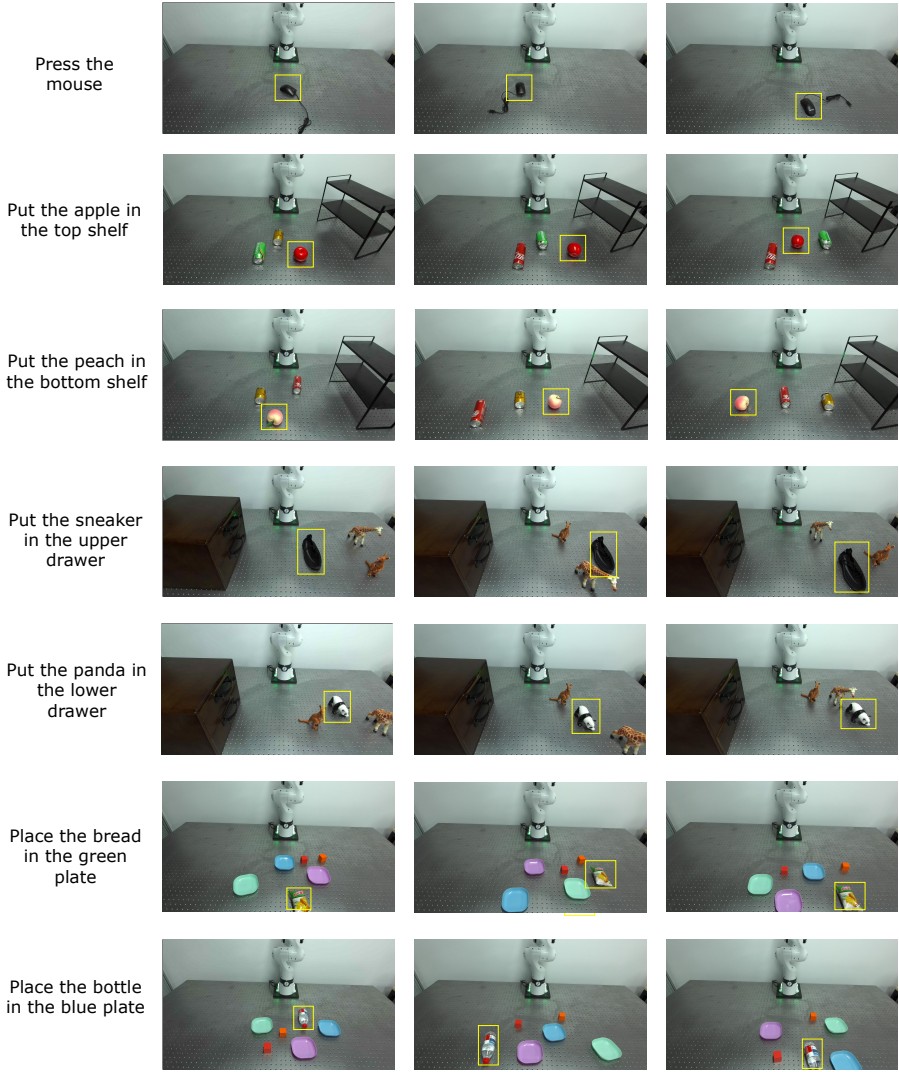

Figure 13: **Visualization of the Category Setting.** In total, we evaluate on 7 objects from novel categories that are unseen during training.

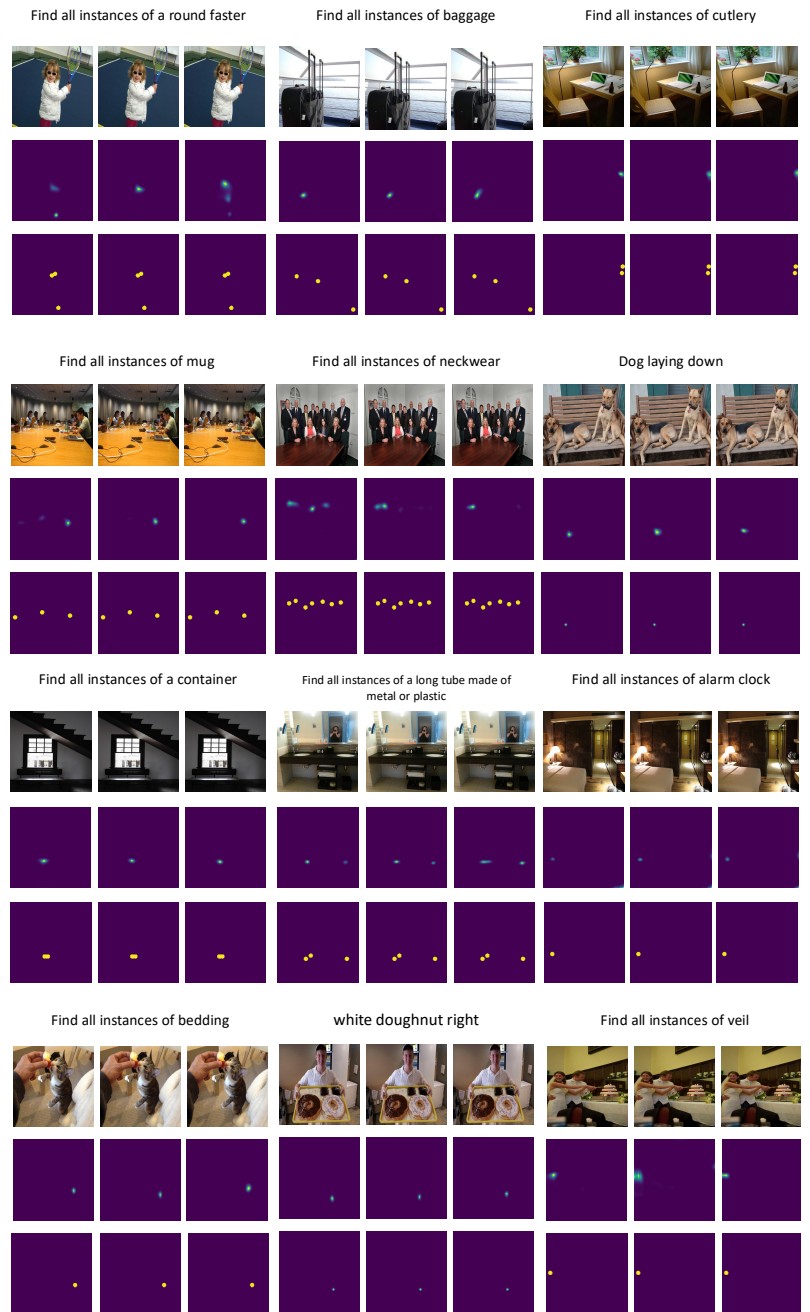

Figure 14: **Visualization of BridgeVLA's Prediction on Pre-training Dataset after Fine-tuning.** To simulate the multi-view inputs during fine-tuning, we repeat the input image three times and feed them into the fine-tuned model to generate heatmaps. For each sample, the first row shows the input image; the second row shows the heatmap prediction; the third row shows the ground truth.

