# OpenReview forum: "BridgeVLA: Input-Output Alignment for Efficient 3D Manipulation Learning with Vision-Language Models"
_NeurIPS.cc/2025/Conference — NeurIPS 2025 poster_

### Official Review · Reviewer_J4oj · 2025-06-09

**Clarity:** 2
**Significance:** 2
**Originality:** 3
**Rating:** 3
**Confidence:** 4

**Summary:**

This paper presents BridgeVLA, a 3D vision-language-action (VLA) model that leverages the structural priors of 3D data (i.e., point clouds) while mitigating a large distributional shift from the original vision-language model training. A key idea of BridgeVLA is to transform 3D point cloud observation into multiple 2D images and further learn to predict a 2D heatmap highlighting regions of interest. 2D heatmaps are used to predict robotic actions. Experiments show that BridgeVLA outperforms several baselines on simulated environments (RLBench and COLOSSEUM) and real-world environments.

**Questions:**

N/A

**Ethical Concerns:**

["NO or VERY MINOR ethics concerns only"]

**Final Justification:**

In the rebuttal phase, the authors addressed my concerns regarding clarity, component-level analysis, and empirical justification. While they have made notable progress, the manuscript still requires significant effort to improve the clarity and depth of analysis of the newly added experiments (e.g., explaining the large performance gap compared to SpatialVLA and π0) before it can meet the bar for publication. Therefore, I am finalizing my rating as borderline reject.

**Limitations:**

yes

**Paper Formatting Concerns:**

No paper formatting concerns found

**Quality:**

2

**Strengths And Weaknesses:**

**Strengths**

- (S1) The manuscript proposes a new 3D VLA model for robotic tasks
- (S2) The paper conducted extensive analyses across real-world and simulation evaluations
- (S3) The paper is generally well-written

**Weaknesses**

- (W1) **Limited empirical justification for core claims**. The current manuscript mainly argues:
  - (1) Current 3D VLA models [1,2] are sample inefficient since they do not fully exploit the spatial structure inherent in 3D data.
  - (2) Using 3D data for fine-tuning models trained on 2D inputs introduces a large distributional shift that harms performance.
- However, the authors did not validate these critical claims in their experiments. They only compared BridgeVLA to non-VLA baselines (e.g., RVT-2), without benchmarking against any existing 3D VLA models. This omission weakens the empirical support for their critique of prior work and leaves the practical advantages of the proposed method unsubstantiated.

- (W2) **Lack of component-level analysis**. The paper claims superior performance over prior work across diverse environments, but it does not provide a deeper investigation into the sources of these improvements. Without detailed ablations or analysis isolating the contributions of individual design choices, it is difficult for the community to interpret where the gains come from or derive actionable insights for future work.

- (W3) **Lack of clarity in core components**. Some technical details of the proposed method are insufficiently described. In Chapter 3, key components of BridgeVLA—such as convex upsampling and the loss functions—are not presented with enough precision. This lack of clarity hinders reproducibility and makes it difficult for readers to fully understand the technical underpinnings of the approach.

**References**
- [1] 3D-VLA: A 3D Vision-Language-Action Generative World Model. ICML 2024.
- [2] SpatialVLA: Exploring Spatial Representations for Visual-Language-Action Model. RSS 2025.

---

> ### Author Rebuttal · Authors · 2025-07-30
>
> Dear Reviewer,
>
> We greatly appreciate your careful review and valuable suggestions. In the following, we offer detailed clarifications and additional analysis in response to your comments, and we hope these address your concerns.
>
> ### **Concern1:  Limited empirical justification for core claims**
>
> We are deeply grateful for your constructive comments. To demonstrate BridgeVLA’s advantages, we compared it with SpatialVLA, a state-of-the-art 3D VLA method. As the word count is limited and a detailed comparison has already been provided in our response to **Concern 1 by Reviewer gMBw**, we avoid repeating it here and kindly refer you to that reply.
>
> ---
>
> ### **Concern 2: Lack of component-level analysis**
>
> Thank you for your insightful comments. Our paper includes ablations on the pretraining technique, showing it greatly improves BridgeVLA’s generalization—especially in the Combination setting (from 5% to 64% success). Your suggestion goes a step further, and we agree that more ablations are valuable for understanding the source of these gains. Below, we present two additional studies, which will be included in the final version.
>
> **1) Whether we need to predict heatmaps before predicting actions**
>
> Our approach avoids direct action prediction by first generating 2D heatmaps using a convex upsampling module. Target positions are then computed by projecting 3D workspace points onto the heatmaps and selecting the point with the highest mean probability.
>
> For ablation, we replaced the convex upsampling module (309M parameters) with a similarly sized Transformer decoder (303M) to directly predict target positions, supervised by MSE loss. All other modules were kept fixed. We performed a hyperparameter grid search and evaluated the model on RLBench. Results are shown in the table below.
>
> Replacing heatmap prediction with direct position regression reduced the average success rate from 88.2% to 31.4%, confirming the effectiveness of our heatmap-based design. The ablated model was also harder to train and more sensitive to hyperparameters—requiring a batch size of 192 and careful learning rate tuning—while our original model trains reliably with a batch size of 64.
>
> We see three main reasons for this outcome: (1) Heatmaps offer denser supervision than 3D position vectors, enabling more effective learning. (2) Projecting 3D points onto heatmaps introduces helpful spatial priors, easing the learning process. (3) The 2D heatmaps share the same spatial structure as the input images, enhancing alignment and improving performance.
>
> **2) Whether we need to remove the 3D position input to the VLM backbone**
>
> Unlike typical 3D VLA models like SpatialVLA, we deliberately **avoid** using per-pixel 3D position inputs and rely solely on RGB images. This design preserves alignment between the input feature spaces of fine-tuning and VLM pretraining, which we find crucial for effective vision-language-action (VLA) modeling.
>
> To test this, we added a 3D convolutional module to encode per-pixel 3D positions, fused them with 2D features, and fed the result into the backbone. Although this adds richer spatial cues, it alters the image feature distribution seen during pretraining, leading to a performance drop from 88.2% to 56.2% on RLBench. Detailed results are shown in the table below.
>
> | Model               | close jar | reach & drag | insert peg | meat grill | open drawer | place cups | place wine | push button | put cupboard |
> |---------------------|-----------|--------------|------------|------------|--------------|-------------|-------------|--------------|----------------|
> | BridgeVLA w/o heat  | 49.3±2.3  | 65.3±2.3     | 0.0±0.0    | 81.3±4.6   | 74.7±10.1    | 1.3±2.3     | 32.0±14.4   | 54.7±6.1     | 5.3±2.3       |
> | BridgeVLA w pos     | 96.0±0.0  | 58.7±6.1     | 26.7±2.3   | 96.0±0.0   | 97.3±2.3     | 14.7±4.6    | 81.3±8.3    | 86.7±2.3     | 10.7±2.3      |
> | BridgeVLA           | **100.0±0.0** | **100.0±0.0**    | **88.0±2.8**   | **100.0±0.0**  | **100.0±0.0**    | **58.4±10.0**   | **88.0±2.8**   | **98.4±2.2**     | **73.6±4.6**      |
>
> | Model               | put drawer | put money  | light bulb | slide block | place shape | stack blocks | stack cups | sweep dustpan | turn tap  | Average |
> |---------------------|------------|------------|-------------|--------------|--------------|---------------|-------------|----------------|------------|---------|
> | BridgeVLA w/o heat  | 0.0±0.0    | 58.7±22.7  | 2.7±2.3     | 64.0±0.0     | 4.0±4.0     | 0.0±0.0       | 0.0±0.0     | 32.0±4.0       | 40.0±10.6 | 31.4    |
> | BridgeVLA w pos     | 78.7±2.3   | 97.3±4.6   | 16.0±4.0    | 72.0±0.0     | 21.3±8.3    | 17.3±2.3      | 4.0±4.0     | 53.3±2.3       | 84.0±0.0  | 56.2    |
> | BridgeVLA           | **99.2±1.8**   | **99.2±1.8**   | **87.2±6.6**    | **96.0±2.8**     | **60.8±7.7**    | **76.8±8.7**      | **81.6±3.6**   | **87.2±1.8**       | **92.8±3.3**  | **88.2**    |
>
> ---
>
> ### **Concern3:  Lack of clarity in core components.**
>
> We apologize for the confusion caused by our insufficient explanation. We will add a more detailed description to the Method section in the final version. As the full implementation is available on the anonymous project site and will be open-sourced upon acceptance, we believe this does not significantly affect reproducibility.
>
> Below, we provide a more detailed explanation of the two components you mentioned:
>
>
> ### 1) Convex Upsampling Module
>
> This module performs adaptive, content-aware interpolation to upsample low-resolution features into high-resolution outputs. Unlike fixed methods (e.g., bilinear or nearest-neighbor), it learns pixel-wise interpolation weights, allowing for finer spatial detail recovery.
>
> It consists of two parallel sub-networks:
>
> - **Output Network (`net_out`)**: A 3-layer convolutional network
>   `in_dim → 2×in_dim → 2×in_dim → out_dim`,
>   which generates low-resolution output features.
>
> - **Mask Network (`net_mask`)**: A 2-layer convolutional network that predicts interpolation weights with output dimension
>   `(up_ratio² × up_kernel²)`. These weights are normalized using **softmax** to form a valid convex combination, ensuring stability and interpretability.
>
> The pseudocode of the forward function is as follows:
>
> ```python
> def forward(self, x):
>     """
>     :param x: (bs, in_dim, h, w)
>     :return: (bs, out_dim, h*up_ratio, w*up_ratio)
>     """
>     bs, c, h, w = x.shape
>     assert c == self.in_dim, c
>     out_low = self.net_out(x)  # low resolution output
>
>     mask = self.mask_scale * self.net_mask(x)
>     mask = mask.view(bs, 1, self.up_kernel**2, self.up_ratio, self.up_ratio, h, w)
>     mask = torch.softmax(mask, dim=2)
>
>     out = F.unfold(
>         out_low,
>         kernel_size=[self.up_kernel, self.up_kernel],
>         padding=self.up_kernel // 2,
>     )
>     out = out.view(bs, self.out_dim, self.up_kernel**2, 1, 1, h, w)
>
>     out = torch.sum(out * mask, dim=2)
>     out = out.permute(0, 1, 4, 2, 5, 3)
>     out = out.reshape(bs, self.out_dim, h * self.up_ratio, w * self.up_ratio)
>
>     return out
> ```
>
> The entire process is **end-to-end differentiable**, making it suitable for joint training with the rest of the model. This module is adapted from the **RAFT** paper [1], and we encourage reviewers to refer to that work for further technical details.
>
> ### 2）Loss function
>
> **For pre-training**, the heatmap prediction loss $\mathcal{L}\_{\text{heat}} $is the only supervision signal. Therefore, the total pre-training loss is:
> $$ \mathcal{L}\_{\text{p}} = \mathcal{L}\_{\text{heat}} $$
>
> Since the heatmap represents a probability distribution over 2D space, we use cross-entropy loss to measure the difference between the predicted and ground-truth heatmaps. Let $P$ be the ground-truth heatmap distribution, $Q$ the predicted distribution, $N$ the number of heatmaps, and $H$,$ W$ the height and width of each heatmap. Then:
> $$ \mathcal{L}\_{\text{heat}} = - \sum\_{n=1}^{N} \sum\_{i=1}^{H} \sum\_{j=1}^{W} P[i, j] \cdot \log Q[i, j] $$
>
> **For finetuning**, the total fine-tuning loss $\mathcal{L}\_f$ is composed of four parts:
> $$ \mathcal{L}\_f = \mathcal{L}\_{\text{heat}} + \mathcal{L}\_{\text{rot}} + \mathcal{L}\_{\text{grip}} + \mathcal{L}\_{\text{col}} $$
>
> - **$\mathcal{L}\_{\text{heat}}$**: Same as in pre-training. Since we compute the target position from the heatmap, this loss can supervise position prediction.
>
> - **$\mathcal{L}\_{\text{rot}}$**: Supervises the predicted end-effector rotation. We discretize each Euler angle (x, y, z) into 72 bins and use cross-entropy loss. Let $x, y, z $denote the ground-truth distributions, and $\hat{x}, \hat{y}, \hat{z} $ the predicted distributions. Then:
> $$ \mathcal{L}\_{\text{rot}} = - \sum\_{i=1}^{72} \left( x[i] \cdot \log \hat{x}[i] + y[i] \cdot \log \hat{y}[i] + z[i] \cdot \log \hat{z}[i] \right) $$
>
> - **$\mathcal{L}\_{\text{grip}}$**: Supervises the gripper open/close status (binary classification). Let $z\_{\text{grip}} \in \{0, 1\}$ be the ground-truth label, and $\hat{z}\_{\text{grip}} \in [0, 1]$ the predicted probability of the gripper being open. Then:
> $$ \mathcal{L}\_{\text{grip}} = - \left( z\_{\text{grip}} \cdot \log \hat{z}\_{\text{grip}} + (1 - z\_{\text{grip}}) \cdot \log (1 - \hat{z}\_{\text{grip}}) \right) $$
>
> - **$\mathcal{L}\_{\text{col}}$**: Supervises whether the predicted motion avoids collision (binary classification). Let $z\_{\text{col}} \in \{0, 1\}$ be the ground-truth label, and $\hat{z}\_{\text{col}} \in [0, 1]$ the predicted probability of avoiding collision. Then:
> $$ \mathcal{L}\_{\text{col}} = - \left( z\_{\text{col}} \cdot \log \hat{z}\_{\text{col}} + (1 - z\_{\text{col}}) \cdot \log (1 - \hat{z}\_{\text{col}}) \right) $$
>
> [1] Zachary Teed and Jia Deng. Raft: Recurrent all-pairs field transforms for optical flow. In Computer Vision–ECCV 2020: 16th European Conference, Glasgow, UK, August 23–28, 2020, Proceedings, Part II 16, pages 402–419. Springer, 2020.

---

> ### Author Response · Authors · 2025-08-04
> **Appreciate Any Further Thoughts on Our Response**
>
> Dear Reviewer,
>
> We sincerely thank you for your time and effort in reviewing our paper. We have done our best to address your concerns regarding:
>
> 1. Limited empirical justification for core claims
> 2. Lack of component-level analysis
> 3. Lack of clarity in core components
>
> If you have any additional concerns or questions, please don’t hesitate to let us know — we’d be happy to clarify further.

---

> > ### Comment · Reviewer_J4oj · 2025-08-05
> >
> > I would like to thank the authors for clarifying the proposed method and providing additional analyses. The rebuttal addresses most of my concerns. However, the response to W1 (comparison to other 3D VLA methods) is still a bit preliminary, as it only evaluates 4 use cases and focuses on low-data conditions. I have no further questions at this point and will raise my score accordingly. Thank you for taking the time to address my concerns!

---

> ### Author Response · Authors · 2025-08-06
> **Response to the remaining concerns**
>
> Dear Reviewer,
>
> Thank you for your thoughtful feedback. We’re glad our previous response addressed most of your concerns. Below, we address your remaining two points on the 3D VLA comparison.
>
> ---
> ### Concern 1: Limited comparison with baselines on only 4 tasks.
> We agree that a broader evaluation is necessary to more convincingly demonstrate the advantages of BridgeVLA over other baselines. Therefore, we have expanded our evaluation to include an additional 9 real-world tasks, resulting in a total of 13 tasks—matching the set used in the main paper. All the following baseline models have been re-trained and re-evaluated on these 13 tasks. The updated comparison table is shown below:
>
> | Method         | Put the soda can in the bottom shelf | Put the giraffe in the lower drawer | Place the red block in the blue plate | Press Sanitizer | Put the RedBull can in the top shelf | Put the RedBull can in the bottom shelf | Put the coke can in the top shelf |
> |----------------|--------------------------------------|-------------------------------------|---------------------------------------|------------------|----------------------------------------|------------------------------------------|-----------------------------------|
> | SpatialVLA-50  | 1/10                                 | 1/10                                | 5/10                                  | 6/10             | 3/10                                   | 1/10                                     | 2/10                              |
> | SpatialVLA-10  | 0/10                                 | 0/10                                | 0/10                                  | 2/10             | 0/10                                   | 0/10                                     | 0/10                              |
> | BridgeVLA      | 9/10                                 | 9/10                                | 10/10                                 | 10/10            | 10/10                                  | 10/10                                    | 10/10                             |
>
> | Method         | Place the orange block in the green plate | Place the red block in the purple plate | Place the yellow block in the green plate | Put the zebra in the upper drawer | Put the zebra in the lower drawer | Put the wolf in the upper drawer | Average |
> |----------------|--------------------------------------------|------------------------------------------|--------------------------------------------|-----------------------------------|----------------------------------|-----------------------------------|---------|
> | SpatialVLA-50  | 6/10                                       | 3/10                                     | 5/10                                       | 2/10                              | 0/10                             | 2/10                              | 28.5%   |
> | SpatialVLA-10  | 1/10                                       | 1/10                                     | 0/10                                       | 0/10                              | 0/10                             | 0/10                              | 3.1%    |
> | BridgeVLA      | 10/10                                      | 10/10                                    | 10/10                                      | 9/10                              | 10/10                            | 9/10                              | 96.9%   |
>
> BridgeVLA consistently outperforms SpatialVLA across all tasks. We observe that SpatialVLA exhibits a decline in performance when trained on multiple similar tasks, occasionally struggling with instruction-following—for instance, it often moves to the blue plate regardless of whether the instruction specifies the purple or blue plate. In contrast, BridgeVLA reliably follows instructions and achieves nearly 100% success across all tasks.
>
> ---
>
> ### Concern 2: Evaluation only in low-data settings.
>
> We intentionally focus on low-data conditions for the following reasons:
>
> 1. The central claim of our paper is that BridgeVLA is a data-efficient model capable of performing well across various settings even with limited training data. Therefore, **we believe the current experimental setup have sufficiently supported our claims.**
>
> 2. Conducting experiments in high-data regimes would require collecting a substantially larger dataset, re-training all baseline models, and conducting comprehensive re-evaluations. Due to the limited time available during the rebuttal phase, we were unable to complete these additional experiments. We kindly ask for your understanding in this regard.
>
> ---
> We hope these additions and clarifications address your remaining concerns. Thank you once again for your valuable feedback. If you have any further questions, we would be happy to discuss them with you.

---

> > ### Author Response · Authors · 2025-08-06
> > **Inclusion of ACT and π0 Results in the 13-Task Setting**
> >
> > Dear Reviewer,
> >
> > We have additionally included the evaluation results of ACT and π0 on our updated set of 13 tasks. The comparison table is provided below.
> >
> > | Method         | Put the soda can in the bottom shelf | Put the giraffe in the lower drawer | Place the red block in the blue plate | Press Sanitizer | Put the RedBull can in the top shelf | Put the RedBull can in the bottom shelf | Put the coke can in the top shelf |
> > |----------------|--------------------------------------|-------------------------------------|---------------------------------------|------------------|----------------------------------------|------------------------------------------|-----------------------------------|
> > | SpatialVLA-50  | 1/10                                 | 1/10                                | 5/10                                  | 6/10             | 3/10                                   | 1/10                                     | 2/10                              |
> > | SpatialVLA-10  | 0/10                                 | 0/10                                | 0/10                                  | 2/10             | 0/10                                   | 0/10                                     | 0/10                              |
> > | π0             | 0/10                                 | 0/10                                | 2/10                                  | 1/10             | 0/10                                   | 1/10                                     | 0/10                              |
> > | ACT            | 2/10                                 | 2/10                                | 3/10                                  | 2/10             | 3/10                                   | 1/10                                     | 2/10                              |
> > | BridgeVLA      | 9/10                                 | 9/10                                | 10/10                                 | 10/10            | 10/10                                  | 10/10                                    | 10/10                             |
> >
> > | Method         | Place the orange block in the green plate | Place the red block in the purple plate | Place the yellow block in the green plate | Put the zebra in the upper drawer | Put the zebra in the lower drawer | Put the wolf in the upper drawer | Average |
> > |----------------|--------------------------------------------|------------------------------------------|--------------------------------------------|-----------------------------------|----------------------------------|-----------------------------------|---------|
> > | SpatialVLA-50  | 6/10                                       | 3/10                                     | 5/10                                       | 2/10                              | 0/10                             | 2/10                              | 28.5%   |
> > | SpatialVLA-10  | 1/10                                       | 1/10                                     | 0/10                                       | 0/10                              | 0/10                             | 0/10                              | 3.1%    |
> > | π0             | 0/10                                       | 0/10                                     | 1/10                                       | 0/10                              | 0/10                             | 0/10                              | 3.8%    |
> > | ACT            | 2/10                                       | 3/10                                     | 4/10                                       | 1/10                              | 2/10                             | 2/10                              | 22.3%   |
> > | BridgeVLA      | 10/10                                      | 10/10                                    | 10/10                                      | 9/10                              | 10/10                            | 9/10                              | 96.9%   |
> >
> >
> > Consistent with the 4-task setting, both ACT and π0 continue to lag significantly behind BridgeVLA across all tasks.

---

### Official Review · Reviewer_Qejn · 2025-06-29

**Clarity:** 3
**Significance:** 3
**Originality:** 2
**Rating:** 4
**Confidence:** 3

**Summary:**

BridgeVLA is a 3D vision-language-action (VLA) model that addresses the misalignment between 3D inputs and 2D pre-trained vision-language models(VLMs). BridgeVLA first pre-trains a vision-language models to predict 2D object heatmaps on standard images, then fine-tunes it on 3D robot data by rendering RGB-D point clouds into three orthographic views and using the resulting heatmaps to drive end-effector translation, rotation, gripper, and collision predictions. This approach not only reconciles the 3D-2D modality gap but also outperforms prior 3D manipulation methods - achieving higher success rates on RLBench benchmarks.

**Questions:**

The questions below extend the weakness outlined above:

1. Prior work (e.g., HAMSTER[1]) already pre-trains on 2D images and then uses a 3D policy. How does BridgeVLA differ from HAMSTER?
2. HAMSTER's([1]) 2D pre-training also includes a task for the VLM to predict object pixel locations in the image. Compared to that, what makes BridgeVLA's heatmap pre-training more spatially aware?
3. In Table 1, can BridgeVLA be directly compared to other VLA methods on RLBench? Could you show a head-to-head comparison where OpenVLA[2], π₀[3], HAMSTER[1], etc., are fine-tuned using the same amount of robot data?
4. Inference seems potentially slow - what is BridgeVLA's actual inference time (in HZ)?

**Ethical Concerns:**

["NO or VERY MINOR ethics concerns only"]

**Final Justification:**

Most of my concerns and questions have been addressed.

**Limitations:**

yes

**Quality:**

2

**Strengths And Weaknesses:**

Strengths:

1. Unified 2D Alignment : By learning a shared 2D heatmap representation in both pre-training and fine-tuning, BridgeVLA aligns 2D features with 3D robot data.
2. Data Efficiency : BridgeVLA achieves strong task performance with very limited robot demonstrations (3 trajectories per task)
3. Robustness & Generalization : BridgeVLA demonstrates resilience under visual disturbances and generalizes to novel objects and unseen object-skill combinations.

Weakness:

1. No Direct Baseline Comparison : The paper does not report head-to-head performance against other VLA methods (e.g., OpenVLA[2], π0[3], HAMSTER[1]).
2. Limited Novelty : Prior works(HAMSTER[1]) already pre-train vision-language models on 2D images to learn spatial representations that a downstream 3D policy can directly leverage, and some even train the VLM to predict end-effector trajectories in the image plane. BridgeVLA's main technical departures are its specific 2D heatmap - based pre-training architecture, its orthographic projection and the addition of a motion planner after action prediction. To make the contribution clearer, the authors should demonstrate how their approach differs from prior works([1],[2],[3]) and pinpoint which components-such as heatmap prediction, orthographic projections, or the post-prediction motion planner-specifically drive the observed performance improvements.
3. Unreported Inference Speed : Inference latency isn't measured. Real-time applicability remains underexplored.

[1] Li, Yi, et al. "Hamster: Hierarchical action models for open-world robot manipulation."arXiv preprint arXiv:2502.05485 (2025).

[2] Kim, Moo Jin, et al. "Openvla: An open-source vision-language-action model."arXiv preprint arXiv:2406.09246 (2024).

[3] Black, Kevin, et al. "π0: A vision-language-action flow model for general robot control, 2024."URL[https://arxiv](https://arxiv/). org/abs/2410.24164

---

> ### Author Rebuttal · Authors · 2025-07-31
>
> Dear Reviewer,
>
> Thank you very much for your time and thoughtful feedback. Below, we provide further clarifications and additional analysis in response to your concerns, and we hope our responses address them satisfactorily.
>
> ### **Concern1: Difference between HAMSTER and BridgeVLA**
>
> Thank you for your constructive comments. We completely agree that HAMSTER shares similarities with BridgeVLA as they both pretrain on 2D images. However, they also differ in several key aspects:
>
> **Architecture Design:**
> - HAMSTER adopts a **hierarchical architecture**, where a frozen VLM generates 2D image-space paths, which are then interpreted by a separate low-level 3D policy to output actions.
> - BridgeVLA is a fully **end-to-end** model that uses a VLM as the backbone. All components, including the VLM, are trained jointly using action supervision.
>
> **Training Strategy:**
> - In HAMSTER, the vision-language model (VLM) is frozen during finetuning, and the action loss only updates the low-level manipulation policy. Additionally, its VLM pretraining predicts object pixel locations by autoregressively generating language token sequences. These sequences are temporally ordered but lack intrinsic spatial structure.
> - In BridgeVLA, it adopts an end-to-end training strategy where the action loss updates all model parameters, including those of the VLM. Its heatmap pretraining directly produces 2D heatmaps with explicit spatial structure and each heatmap pixel corresponds one-to-one with a pixel in the input image.
>
> **Compounding error:**
> - HAMSTER is prone to compounding error, as mistakes in either the VLM’s path prediction or the low-level policy can cause task failure.
> - BridgeVLA avoids such compounding error as heatmap-to-position process uses a deterministic projection from 3D points to image planes and selects the point with the highest probability on the heatmap. This step is precise and introduces no learning-related uncertainty.
>
> **Integration Possibility:**
> - BridgeVLA can serve as an improved low-level policy within HAMSTER’s hierarchy. It transforms point clouds into multi-view images, on which we can draw image paths using HAMSTER’s pretrained VLM.
> - In HAMSTER’s original implementation, RVT-2 and 3D Diffuser Actor are used as the low-level policy. As shown in our experiments, BridgeVLA significantly outperforms both, and we believe its integration could further enhance HAMSTER's performance.
>
> ---
>
> ### **Concern2: What makes BridgeVLA's heatmap pretraining more spatially aware than HAMSTER's VLM pre-training?**
>
> Thank you for your question. We think it is very insightful and important.
>
> HAMSTER's VLM pretraining predicts object pixel locations through autoregressive generation of language token sequences. Although temporally ordered, these sequences lack intrinsic spatial structure. The predicted tokens must be decoded into numeric symbols before they can be associated with pixel coordinates, introducing additional abstraction that weakens spatial grounding.
>
> In contrast, BridgeVLA's heatmap pretraining predicts 2D heatmaps that maintain explicit spatial structure. Each heatmap pixel corresponds one-to-one with a pixel in the input image, enabling strong spatial alignment. This direct spatial correspondence allows BridgeVLA to learn more spatially grounded representations, making it more spatially aware than HAMSTER’s VLM pretraining.
>
> ---
>
> ### **Concern3: Comparison with other VLA methods**
>
> We greatly appreciate your constructive comments. It aligns closely with **Concern 1 raised by Reviewer gMBw**, to which we have provided a detailed response earlier. Due to word limit, we avoid repeating this comparing experiments here. Please kindly refer to our response under Concern 1 raised by Reviewer gMBw.
>
> ---
>
> ### **Concern4: Inference time of BridgeVLA**
> Thank you for your constructive comments. We evaluated the inference speed of BridgeVLA by running it 100 times on a machine equipped with an NVIDIA RTX 4090 GPU. The average end-to-end inference time—from point cloud input to action output—is 0.21 seconds, corresponding to approximately 5 Hz.
>
> ---
>
> ### **Concern5: Ablations on BirdgeVLA's main components**
>
> We are deeply grateful for your constructive comments. In our paper, we conducted ablation studies on pretraining technique. Our real-robot experiments demonstrate that our pretraining significantly improves BridgeVLA's generalization ability, especially under the Combination setting, where the average success rate increases from 5% to 64%. Below, we provide additional clarification regarding the three components you mentioned and we will also add these experiments into our final version.
>
> **1) Heatmap Prediction**
> In our original approach, we avoid direct action prediction by first generating 2D heatmaps through a convex upsampling module. We then compute the target positions by projecting the 3D points in the workspace onto the three heatmaps and selecting the point with the highest mean probability.
>
> For the ablation, we replaced the convex upsampling module (309M parameters) with a Transformer decoder of similar size (303M parameters) to directly predict target positions, and supervise this prediction by MSE loss. The other modules are keep fixed. We conducted hyperparameter grid search for optimal performance, and evaluated the ablated model on RLBench. The results are summarized in Table below.
>
> As shown, replacing heatmap prediction with direct position prediction reduces the average success rate from 88.2% to 31.4%, confirming the effectiveness of our heatmap-based design. Additionally, we found the ablated model was much harder to train and more sensitive to hyperparameters, requiring a batch size of 192 and careful learning rate tuning for training stability, whereas our original model converges reliably even with a batch size of 64.
>
> We believe there are three main reasons for this outcome. First, heatmaps provide a denser supervision signal compared to three-dimensional position vectors, which facilitates more effective learning. Second, the process of deriving target positions through projecting 3D points onto heatmaps implicitly incorporates additional spatial priors into the pipeline, thereby reducing the learning difficulty for the model. Third, the 2D heatmap outputs are defined in the same spatial domain as the 2D image inputs. This spatial alignment helps the model better capture the correspondence between input and output, ultimately leading to improved performance.
>
> **2) Orthographic Projection**
> It is not feasible to ablate orthographic projection in isolation. Without multi-view projections, we cannot compute 3D target positions from 2D heatmaps, and the entire pipeline would fail.
>
> Meanwhile, orthographic projection itself is not the main focus—it is primarily a tool we use to convert 3D point clouds into 2D images. This transformation ensures that the input feature space during fine-tuning is well aligned with the VLM pretraining space, which we believe is crucial for building a unified vision-language-action (VLA) model.
>
> To test the importance of this alignment, we conducted an additional ablation inspired by SpatialVLA. Specifically, we added a 3D convolutional module to encode 3D positions of each image patch, which were then fused with the corresponding 2D image features before feeding into the backbone. While this introduces richer spatial information, it significantly shifts the distribution of image features compared to those seen during VLM pretraining. As a result, although we did hyperparameter grid search to get optimal performance, it still dropped from 88.2% to 56.2% on RLBench. Detailed results are shown in Table below.
>
> **3) Motion Planner**
> BridgeVLA is a keypoint-based method, and all such methods have to rely on a motion planner to move between predicted keypoints. As such, this component can not be regarded as our contribution and cannot be meaningfully ablated in isolation.
> In our experiments, we simply use default motion planners—RRT-Connect. If you had a specific type of ablation in mind regarding this module, we would greatly appreciate it if you could clarify your expectations.
>
> | Model               | close jar | reach & drag | insert peg | meat grill | open drawer | place cups | place wine | push button | put cupboard |
> |---------------------|-----------|--------------|------------|------------|--------------|-------------|-------------|--------------|----------------|
> | BridgeVLA w/o heat  | 49.3±2.3  | 65.3±2.3     | 0.0±0.0    | 81.3±4.6   | 74.7±10.1    | 1.3±2.3     | 32.0±14.4   | 54.7±6.1     | 5.3±2.3       |
> | BridgeVLA w pos     | 96.0±0.0  | 58.7±6.1     | 26.7±2.3   | 96.0±0.0   | 97.3±2.3     | 14.7±4.6    | 81.3±8.3    | 86.7±2.3     | 10.7±2.3      |
> | BridgeVLA           | **100.0±0.0** | **100.0±0.0**    | **88.0±2.8**   | **100.0±0.0**  | **100.0±0.0**    | **58.4±10.0**   | **88.0±2.8**   | **98.4±2.2**     | **73.6±4.6**      |
>
> | Model               | put drawer | put money  | light bulb | slide block | place shape | stack blocks | stack cups | sweep dustpan | turn tap  | Average |
> |---------------------|------------|------------|-------------|--------------|--------------|---------------|-------------|----------------|------------|---------|
> | BridgeVLA w/o heat  | 0.0±0.0    | 58.7±22.7  | 2.7±2.3     | 64.0±0.0     | 4.0±4.0     | 0.0±0.0       | 0.0±0.0     | 32.0±4.0       | 40.0±10.6 | 31.4    |
> | BridgeVLA w pos     | 78.7±2.3   | 97.3±4.6   | 16.0±4.0    | 72.0±0.0     | 21.3±8.3    | 17.3±2.3      | 4.0±4.0     | 53.3±2.3       | 84.0±0.0  | 56.2    |
> | BridgeVLA           | **99.2±1.8**   | **99.2±1.8**   | **87.2±6.6**    | **96.0±2.8**     | **60.8±7.7**    | **76.8±8.7**      | **81.6±3.6**   | **87.2±1.8**       | **92.8±3.3**  | **88.2**    |

---

> > ### Comment · Reviewer_Qejn · 2025-08-05
> >
> > Thank you to the authors for the clarification. Most of my concerns and questions have been addressed. I have raised my score.

---

> ### Author Response · Authors · 2025-08-04
> **Appreciate Any Further Thoughts on Our Response**
>
> Dear Reviewer,
>
> We sincerely thank you for your time and effort in reviewing our paper. We have done our best to address your concerns regarding:
>
> 1. Difference between HAMSTER and BridgeVLA
> 2. What makes BridgeVLA's heatmap pretraining more spatially aware than HAMSTER's VLM pre-training
> 3. Comparison with other VLA methods
> 4. Inference time of BridgeVLA
> 5. Ablations on BirdgeVLA's main components
>
> If you have any additional concerns or questions, please don’t hesitate to let us know — we’d be happy to clarify further.

---

### Official Review · Reviewer_gMBw · 2025-07-03

**Clarity:** 3
**Significance:** 4
**Originality:** 4
**Rating:** 5
**Confidence:** 5

**Summary:**

The paper proposed, BridgeVLA a 3D vision-language-action (VLA) model that projects 3D inputs into multiple 2D images and uses 2D heatmaps for action prediction, aligning both input and output spaces for efficient robot manipulation learning. The authors train the model in 2 stages:

a) pre-training the VLM backbone to predict heatmaps with task relevant points, and
b) training a policy on top of the heat maps to predict the end-effector pose of the robot.

BridgeVLA achieves state-of-the-art results on the RL-Bench benchmark and shows significant robustness on the COLLESUM benchmark.

**Questions:**

3. What is the inference time for the overall policy?

4. In Table 4, what could be the reason that BridgeVLA performs on "RO-TEXTURE" than on "NO Variations"?

5. There are no long-horizon tasks included in the evaluation (both in sim and real world). I'm wondering if the pipeline would still work when chained with multiple tasks. For instance, "Put the Red Bull can on the bottom shelf and the Coke can on the top shelf." I hypothesize that the heatmap generation may fail (or maybe not) as it wasn't trained on such prompts. But any insights on this would be helpful.

**Ethical Concerns:**

["NO or VERY MINOR ethics concerns only"]

**Final Justification:**

Thanks to the authors for addressing all my concerns (adding VLA baselines, addressing differences between RO-TEXTURE and No-Variations, adding ACT baseline).

I will continue to maintain my score of **Accept**.

**Limitations:**

Yes.

**Paper Formatting Concerns:**

None.

**Quality:**

3

**Strengths And Weaknesses:**

**Strengths**

- The method is sound and novel and furthers the boundary of 2D heatmap-based robotic manipulation.
- Leveraging the RoboPoint dataset for pre-training is a very neat way of learning 2D heatmap priors for the downstream task of manipulation.
- Well-written paper.


**Weaknesses**:

1. Comparison to alternative 3D VLA method like FP3 is missing. I'm curious if the heatmap-based approaches perform significantly better than natively trained 3D policies. (PointVLA doesn't open-source their code, so a comparison there isn't required).

2. ACT [1] has become a strong BC baseline in recent years, so I believe that having the ACT baseline the way the Diffusion Policy baseline (3D diffuser actor) was added would make this paper stronger.

----

**References**

[1] Learning Fine-Grained Bimanual Manipulation with Low-Cost Hardware, Tony Zhao et al., RSS 2023

---

> ### Author Rebuttal · Authors · 2025-07-31
>
> Dear Reviewer,
>
> Thank you very much for your valuable feedback. We address your concerns with the following clarifications and additional analysis, and hope our responses are helpful.
>
> ### **Concern1: Comparison to other 3D VLA methods**
>
> We sincerely thank you for the constructive feedback. In response, we conducted additional comparisons between BridgeVLA and three types of manipulation policies, including SpatialVLA as a 3D VLA method. These results will be included in the final version of the paper.
>
> 1）SpatialVLA: A state-of-the-art **3D VLA** model that incorporates 3D information through Ego3D positional encoding and leverages Adaptive Action Grids to accelerate inference.
>
> 2）π0: A state-of-the-art **2D VLA** model pretrained on a large-scale cross-embodiment dataset. It adopts a vision-language model (VLM) backbone and employs a diffusion-based action expert to generate final actions.
>
> 3）ACT:  A state-of-the-art 2D **non-VLA** model using a Conditional Variational Autoencoder (CVAE) to model action distributions. Though effective for fine-grained manipulation, ACT does not support language conditioning, so we train a separate single-task model for each task, which should theoretically perform better than a multi-task version.
>
> As for **3D non-VLA** models, we have already compared with leading methods such as RVT-2 and 3D Diffuser Actor in the paper, and thus do not repeat them here.
>
> Due to time constraints, we were unable to test all 13 real-robot tasks from the paper. Since some tasks are similar (e.g., "Put the RedBull can in the bottom shelf" vs. "Put the soda can in the bottom shelf"), we selected four representative tasks for comparison. As in the original setting, each baseline was trained with 10 trajectories per task and evaluated over 10 trials to ensure statistical robustness. For fair comparison, we photographed each test scene and manually aligned the setups across all methods.
>
> The results are presented below, along with our observations:
>
> | Model           | Put the soda can in the bottom shelf | Put the giraffe in the lower drawer | Place the red block in the blue plate | Press Sanitizer | Average |
> |----------------|--------------------------------------|--------------------------------------|----------------------------------------|------------------|---------|
> | SpatialVLA-50  | 2/10                                 | 2/10                                 | 4/10                                   | 6/10             | 35.0%   |
> | SpatialVLA-10  | 0/10                                 | 0/10                                 | 0/10                                   | 1/10             | 2.5%    |
> | π0             | 0/10                                 | 0/10                                 | 1/10                                   | 1/10             | 5.0%    |
> | ACT            | 1/10                                 | 2/10                                 | 3/10                                   | 3/10             | 22.5%   |
> | BridgeVLA      | **10/10**                                | **9/10**                                 | **10/10**                                  | **10/10**            | **97.5%**   |
>
>
>
> **SpatialVLA**: According to the experimental setup, we initially trained SpatialVLA with only 10 trajectories per task. However, it failed on nearly all tasks, often struggling even to move toward the correct target object. Therefore, we augmented the dataset with an additional 40 trajectories per task. The performance improved but remained substantially lower than that of BridgeVLA—particularly on more challenging tasks such as *Put the giraffe in the lower drawer*. These findings suggest that BridgeVLA offers a more effective and data-efficient 3D VLA solution.
>
> **π0**: Similarly, π0 fails with only 10 trajectories per task, likely due to overfitting—it performs well on training inputs but often fails during online testing. Common failure modes include missing or failing to grasp the target, and prematurely opening the gripper before reaching the goal. Notably, both BridgeVLA and π0 share the same PaliGemma backbone and are trained end-to-end. This underscores a key contribution of our work: while VLAs like π0 perform well with large-scale data, **they struggle in low-data regimes**—even on simple tasks like Press sanitizer. In contrast, BridgeVLA achieves near-perfect success and generalizes robustly across diverse settings.
>
> **ACT**: ACT also underperforms compared to BridgeVLA. It shows limited spatial generalization—performing well only in areas densely covered during training, but often failing when the target is near workspace boundaries. This aligns with its design: ACT models actions with a Gaussian prior, which assigns low probability to peripheral regions, limiting spatial generalization.
>
> Given that none of these baseline models can reliably solve the basic settings under such low-data conditions, we did not proceed with further evaluations on generalization or transfer scenarios.
>
> ---
>
> ### **Concern2: Comparison with ACT**
>
> Thank you for your insightful suggestion. We have addressed this concern in our response to Concern 1.
>
> ---
>
> ### **Concern3: Inference time for the overall policy**
>
> We greatly appreciate your constructive comments. We evaluated the inference speed of BridgeVLA by running it 100 times on a machine equipped with an NVIDIA RTX 4090 GPU. From point cloud input to action output, the average end-to-end inference time is 0.21 seconds.
>
> ---
>
> ### **Concern4: Why BridgeVLA performs better on "RO-TEXTURE" than on "NO Variations"?**
>
> We sincerely apologize for the confusion caused by our earlier mistake. The original evaluation script contained some minor bugs, which have now been fixed. After correction, BridgeVLA performs  worse on the "RO-TEXTURE" setting (68.4%) compared to the "NO Variations" setting (73.9%).
>
> Importantly, this does not impact the main conclusions, as BridgeVLA still clearly outperforms all baselines on the COLOSSEUM benchmark. The correct success rate is provided below, and we will update the table in the final version.
>
> | Models      | Avg. SR (%) | Avg. Rank | All Perturbations | MO-COLOR | RO-COLOR | MO-TEXTURE | RO-TEXTURE | MO-SIZE |
> |-------------|--------------|------------|---------------------|------------|-------------|---------------|---------------|-----------|
> | R3M-MLP     | 0.8          | 5.71       | 0.6                 | 0.4        | 0.0         | 0.0           | 0.0           | 1.8       |
> | MVP-MLP     | 1.6          | 5.00       | 0.8                 | 1.2        | 0.0         | 0.4           | 0.0           | 4.44      |
> | PerAct      | 27.9         | 3.71       | 7.2                 | 24.0       | 29.2        | 28.8          | 17.71         | 35.6      |
> | RVT         | 35.4         | 3.28       | 6.4                 | 26.0       | 31.3        | 44.8          | 41.1          | 35.3      |
> | RVT-2       | 56.7         | 1.92       | 15.6                | 53.0       | 54.6        | 59.7          | 56.7          | 60.9      |
> | BridgeVLA   | **64.0**     | **1.07**   | **18.7 ± 2.2**      | **60.5 ± 1.1** | **63.8 ± 0.1**  | **63.5 ± 1.5**    | **68.4 ± 3.3**  | **69.3 ± 1.0** |
>
> | Models      | RO-SIZE | Light Color | Table Color | Table Texture | Distractor | Back-Texture | RLBench | Camera Pose |
> |-------------|---------|-------------|--------------|----------------|-------------|----------------|----------|---------------|
> | R3M-MLP     | 0.0     | 1.0         | 1.4          | 0.2            | 1.6         | 1.2            | 2.0      | 0.8           |
> | MVP-MLP     | 0.0     | 1.6         | 1.6          | 1.0            | 3.8         | 2.2            | 2.0      | 2.6           |
> | PerAct      | 29.3    | 29.1        | 30.4         | 23.2           | 27.1        | 33.5           | 39.4     | 36.3          |
> | RVT         | 40.5    | 34.0        | 30.0         | 45.2           | 18.8        | 46.4           | 53.4     | 42.2          |
> | RVT-2       | 53.4    | 58.0        | 62.6         | 56.6           | 60.8        | 68.7           | 68.8     | 64.4          |
> | BridgeVLA   | **61.7 ± 0.8** | **69.7 ± 1.2** | **75.7 ± 0.9**  | **71.3 ± 0.7**     | **51.8 ± 1.5**  | **74.8 ± 1.0**   | **73.1 ± 0.2** | **73.8 ± 0.3**  |
>
> ---
>
> ### **Concern5: Insights on long horizon tasks**
>
> We are deeply grateful for your thoughtful suggestion regarding long-horizon tasks. We evaluated BridgeVLA on the examples you provided and found it currently cannot handle them effectively. However, several interesting patterns emerged.
>
> We ran two instructions 10 times each:
> - “Put the RedBull can on the bottom shelf and then put the Coke can on the top shelf.”
> - “Put the Coke can on the bottom shelf and then put the RedBull can on the top shelf.”
>
> In all 20 trials, BridgeVLA was consistently able to place one can on the shelf successfully. However, after placing the first can, the robot failed to proceed to the second part of the task, instead moving back and forth near the endpoint without further action. This behavior is understandable, as the model was never trained to return and pick up a second object.
>
> Interestingly, we observed that in 12 out of 20 trials, the model placed the correct can on the correct shelf, while in 8 trials, it did not. Regarding the execution order, the model followed the sequence described in the instruction in 11 out of 20 trials.
>
> This suggests BridgeVLA can recognize object entities but struggles with relational reasoning and temporal sequencing. We believe this is due to our 2D heatmap pretraining, which was based on a 120K object detection dataset. While it helps with object recognition, it lacks exposure to higher-level semantics such as object-location mapping or multi-step task structure.
>
> To address this, we plan to explore pretraining strategies using more semantically diverse datasets to improve BridgeVLA’s ability to interpret and execute complex, multi-step instructions.

---

> ### Author Response · Authors · 2025-08-04
> **Appreciate Any Further Thoughts on Our Response**
>
> Dear Reviewer,
>
> We sincerely thank you for your time and effort in reviewing our paper. We have done our best to address your concerns regarding:
>
> 1. Comparison to other 3D VLA methods
> 2. Comparison with ACT
> 3. Inference time for the overall policy
> 4. Why BridgeVLA performs better on "RO-TEXTURE" than on "NO Variations"
> 5. Insights on long horizon tasks
>
> If you have any additional concerns or questions, please don’t hesitate to let us know — we’d be happy to clarify further.

---

> > ### Comment · Reviewer_gMBw · 2025-08-09
> > **Thanks for the rebuttal**
> >
> > [I had already updated this in my Final justification but posting here after realizing that Final Justification is not visible to authors until the decisions are out[
> >
> > Thanks to the authors for addressing all my concerns (adding VLA baselines, addressing differences between RO-TEXTURE and No-Variations, adding ACT baseline).
> >
> > I will continue to maintain my score of **Accept**.

---

### Official Review · Reviewer_Prgj · 2025-07-03

**Clarity:** 4
**Significance:** 2
**Originality:** 3
**Rating:** 2
**Confidence:** 4

**Summary:**

BridgeVLA is a 3D Vision-Language-Action (VLA) model that enhances robot manipulation by efficiently integrating 3D data with pre-trained vision-language models. It projects 3D inputs into 2D images for compatibility with VLMs and uses 2D heatmaps for action prediction, ensuring spatial consistency. A scalable pre-training phase further improves performance. Experiments show BridgeVLA outperforms state-of-the-art methods, achieving higher success rates with better sample efficiency.

**Questions:**

Please refer to the weaknesses section above for details：
 1.Absence of Ablation Studies
 2. Lack of Comparison with State-of-the-Art VLA Models
 3. Insufficient Experimental Validation of Cross-Modal Bridging

**Ethical Concerns:**

["NO or VERY MINOR ethics concerns only"]

**Limitations:**

Yes

**Quality:**

3

**Strengths And Weaknesses:**

1. We would recommend that the authors consider including ablation studies on the 2D heatmap prediction component.
2.  Lack of Comparison with State-of-the-Art VLA Models: The current approach, which involves predicting 2D Heatmaps prior to action generation, is conceptually similar to methods like ECoT. Despite the authors' claim of using 3D point cloud data, the actual input to the VLA appears to be RGB data, warranting a direct comparison with ECoT. Furthermore, the authors should benchmark their method against contemporary VLA algorithms in a real experimental setting. Relevant comparisons include Pi\_0, Co-Act RDT-1B,OpenVLA.OpenVLA-oft, 3D Diffusion Policy, and ACT.Additionally, we hope the authors can explore the model's performance on tasks such as folding clothes and folding pants.
3.  Insufficient Experimental Validation of Cross-Modal Bridging:The paper asserts that the proposed method bridges the gap between 2D image-input Vision-Language Models (VLMs) and 3D inputs. However, most of the algorithms used for comparison do not utilize VLMs as their backbone, thereby failing to adequately demonstrate this claimed advantage.

---

> ### Author Rebuttal · Authors · 2025-07-31
>
> Dear Reviewer，
>
> We sincerely appreciate your detailed and valuable feedback. In response to the concerns raised, we provide clarifications and additional analysis below. Hope they can address your concerns.
>
> ### **Concern1: Lack of ablation studies on heatmap prediction**
>
> In our original approach, we avoid direct action prediction by first generating 2D heatmaps through a convex upsampling module. We then compute the target positions by projecting the 3D points in the workspace onto the three heatmaps and selecting the point with the highest mean probability.
>
> For the ablation, we replaced the convex upsampling module (309M parameters) with a Transformer decoder of similar size (303M parameters) to directly predict target positions, and supervise this prediction by MSE loss. The other modules are keep fixed. We conducted hyperparameter grid search for optimal performance, and evaluated the ablated model on RLBench. The results are summarized below:
>
> | Model               | close jar | reach & drag | insert peg | meat grill | open drawer | place cups | place wine | push button | put cupboard |
> |---------------------|-----------|--------------|------------|------------|--------------|-------------|-------------|--------------|----------------|
> | BridgeVLA w/o heat  | 49.3±2.3  | 65.3±2.3     | 0.0±0.0    | 81.3±4.6   | 74.7±10.1    | 1.3±2.3     | 32.0±14.4   | 54.7±6.1     | 5.3±2.3       |
> | BridgeVLA           | **100.0±0.0** | **100.0±0.0**    | **88.0±2.8**   | **100.0±0.0**  | **100.0±0.0**    | **58.4±10.0**   | **88.0±2.8**   | **98.4±2.2**     | **73.6±4.6**      |
>
> | Model               | put drawer | put money  | light bulb | slide block | place shape | stack blocks | stack cups | sweep dustpan | turn tap  | Average |
> |---------------------|------------|------------|-------------|--------------|--------------|---------------|-------------|----------------|------------|---------|
> | BridgeVLA w/o heat  | 0.0±0.0    | 58.7±22.7  | 2.7±2.3     | 64.0±0.0     | 4.0±4.0     | 0.0±0.0       | 0.0±0.0     | 32.0±4.0       | 40.0±10.6 | 31.4    |
> | BridgeVLA           | **99.2±1.8**   | **99.2±1.8**   | **87.2±6.6**    | **96.0±2.8**     | **60.8±7.7**    | **76.8±8.7**      | **81.6±3.6**   | **87.2±1.8**       | **92.8±3.3**  | **88.2**    |
>
> As shown, replacing heatmap prediction with direct position prediction reduces the average success rate from 88.2% to 31.4%, confirming the effectiveness of our heatmap-based design. Additionally, we found the ablated model was much harder to train and more sensitive to hyperparameters, requiring a batch size of 192 and careful learning rate tuning for training stability, whereas our original model converges reliably even with a batch size of 64.
>
> We believe there are three main reasons for this outcome. First, heatmaps provide a denser supervision signal compared to three-dimensional position vectors, which facilitates more effective learning. Second, the process of deriving target positions through projecting 3D points onto heatmaps implicitly incorporates additional spatial priors into the pipeline, thereby reducing the learning difficulty for the model. Third, the 2D heatmap outputs have the same spatial structure as the 2D image inputs. This spatial alignment helps the model better capture the correspondence between input and output, ultimately leading to improved performance.
>
> ---
>
> ### **Concern2: Comparison with State-of-the-Art VLA Models**
>
> Thank you for your constructive comments. We fully agree that including more relevant baselines would strengthen our conclusions. This concern aligns closely with **Concern 1 raised by Reviewer gMBw**, to which we have provided a detailed response earlier. Due to word limit, we avoid repeating these comparing experiments here. Please kindly refer to our response under Concern 1 raised by Reviewer gMBw.
>
> ---
>
> ### **Concern3: Comparison with ECoT**
>
> Thanks for your valuable suggestion. We have carefully read the ECoT paper and completely agree that ECoT shares similarities with BridgeVLA, as both involve intermediate representations. We will cite and discuss with ECoT paper in the final version of our manuscript. However, we would like to highlight several key differences between the two approaches:
>
> i）ECoT and BridgeVLA are complementary rather than mutually exclusive. ECoT contributes primarily by enabling reasoning before action execution, while the final action output is generated using an existing manipulation policy such as OpenVLA — which could potentially be replaced by BridgeVLA. Conversely, ECoT’s core strategies—such as task decomposition and leveraging foundation models to generate bounding boxes and end-effector positions—could also be integrated into the BridgeVLA framework.
>
> ii) In ECoT, intermediate results—such as sub-instructions and bounding boxes—are fed into a learnable downstream policy to enhance the accuracy of final action prediction. Therefore, these intermediate results are only used to aid other modules. In contrast, for BridgeVLA, once the 2D heatmaps are generated, it projects 3D points from the workspace onto the heatmaps and selects the point with the highest mean probability as the target position. This process is deterministic and non-learnable. Therefore, heatmaps in BridgeVLA are directly used to get target positions rather than aid other modules.
>
> Since we have already compared against π0, which is also a SOTA 2D VLA model, we do not include a direct comparison with ECoT in this version. We consider the integration of ECoT’s techniques with BridgeVLA a promising direction for future work.
>
> ---
>
> ### **Concern4: Performance on tasks such as folding pants**
>
> Thank you for your insightful suggestion. We agree that incorporating experiments on folding clothes or pants would further validate the effectiveness of BridgeVLA. Since we use a single Franka Research 3 arm as our robotic embodiment, we are unable to perform the highly dexterous cloth-folding tasks demonstrated in π0 due to physical limitations. However, we have followed your suggestions and implemented a simplified version on our system—folding a pair of pants first from left to right, and then from front to back.
>
> To evaluate our model’s performance, we define two metrics: **Task Success Rate (TSR)** and **Sub-Task Success Rate (STSR)**. For TSR, a trial is considered successful only if the robot completes both folding actions—left-to-right and front-to-back—within a single execution. For STSR, a partial score is given: 0.5 is awarded if the robot successfully folds the pants from left to right, and another 0.5 if it subsequently folds them from front to back.
>
> We define success criteria as follows:
> - Left-to-right fold: Considered successful if the left side of the pants is placed beyond the midpoint.
> - Front-to-back fold: Considered successful if the front side is placed beyond the midpoint.
>
> The experimental results are shown below:
>
> | Metric | Score  |
> |--------|--------|
> | TSR    | 5/10   |
> | STSR   | 7.5/10 |
>
> We observe that the robot consistently succeeds in folding the pants from left to right. However, it only succeeds about half the time when folding from front to back. This discrepancy arises because the initial shape of the pants matches those seen in the training data, allowing the model to accurately predict the grasping location for the first fold. However, after the initial fold, the deformable nature of the pants combined with the limitations of single-arm manipulation leads to unpredictable object shapes. As a result, the robot often fails to locate or properly grasp the front side of the pants in the second step.
>
> ---
>
> ### **Concern5: Experimental Validation of Cross-Modal Bridging**
>
> We greatly appreciate your valuable comments. As detailed in our response to Concern 2, we have conducted real robot experiments comparing BridgeVLA with SpatialVLA and π0—both of which employ large-scale VLMs as their backbones. Notably, both BridgeVLA and π0 use the same PaliGemma backbone, while SpatialVLA even adopts the more advanced PaliGemma 2 backbone.
>
> Despite this, the experimental results show that BridgeVLA significantly outperforms both π0 and SpatialVLA (97.5% vs. 5% and 2.5% , respectively). This highlights the effectiveness of our bridging design in more fully leveraging the capabilities of 2D vision-language models when the finetuning data is extremely limited.

---

> ### Author Response · Authors · 2025-08-04
> **Appreciate Any Further Thoughts on Our Response**
>
> Dear Reviewer,
>
> We sincerely thank you for your time and effort in reviewing our paper. We have done our best to address your concerns regarding:
>
> 1. Ablation studies on heatmap prediction
> 2. Comparison with state-of-the-art VLA models
> 3. Comparison with ECoT
> 4. Tasks involving folding pants
> 5. Experimental validation of cross-modal bridging
>
> If you have any additional concerns or questions, please don’t hesitate to let us know — we’d be happy to clarify further.

---

### Author Response · Authors · 2025-08-09
**Response to Reviewer Comments**

Dear Reviewer,

We sincerely appreciate the time and effort you dedicated to reviewing our paper. Your insightful comments and constructive suggestions are invaluable to us.

In this work, we propose BridgeVLA, a novel 3D VLA framework that aligns inputs and outputs within a **shared 2D space** during both pre-training and fine-tuning. **This design enables strong data efficiency—achieving a 96.8% success rate using only 3 demonstrations per task—as well as impressive generalization capabilities.**

The effectiveness of this alignment is supported by extensive ablation studies, including BridgeVLA without pre-training, without heatmap, and with positional encoding. Furthermore, comparisons with state-of-the-art methods—including **3D VLA** (SpatialVLA), **3D non-VLA** (RVT-2, 3D DA), **2D VLA** (π0), and **2D non-VLA** (ACT)—further underscore the strength of our approach.

We believe BridgeVLA will contribute meaningfully to the community and inspire further research in this area. We hope you will consider supporting the acceptance of our paper.

Sincerely,

The Authors

---

### Note · Authors · 2025-08-11

We would like to express our sincere gratitude to all the reviewers for their time and efforts in evaluating our paper. However, we would like to highlight a few points to ensure that BridgeVLA's contributions are properly assessed:

1. In this paper, we introduce a novel 3D VLA framework that bridges the gap between VLM and VLA by aligning both the inputs and outputs within a shared 2D space during pre-training and fine-tuning. Our model achieves **state-of-the-art performance on three simulation benchmarks** and demonstrates **excellent data efficiency** (achieving a 96.8% success rate with only 3 trajectories per task), as well as **strong generalization ability in real-world scenarios** (covering factors such as Distractors, Lighting, Background, Combination, Height, and Category).

2. The effectiveness of our design is validated through comprehensive ablation studies, including experiments with BridgeVLA without pre-training, without heatmap, and with positional encoding. Additionally, comparisons with state-of-the-art methods—such as **3D VLA**(SpatialVLA), **3D non-VLA** (RVT-2, 3D DA), **2D VLA** (π0), and **2D non-VLA** (ACT)—further emphasize the robustness of our approach.

3. Reviewer gMBw has confirmed that we have addressed all of their concerns and **expressed absolute confidence that our paper should be accepted.**

4. Reviewer Qejn also provided positive feedback and raised his score.

5. Reviewer **J4oj** mentioned that our rebuttal addressed most of his concerns and **promised to raise the score**. However, the rating has not changed now. We are unsure if he inadvertently overlooked submitting the final score. **We kindly ask the Area Chair to remind the reviewer of this**. Furthermore, the reviewer raised two additional concerns during the discussion phase. We have since provided additional experimental results and clarifications to address these points, but unfortunately, we have not received any further feedback.

6. **Reviewer Prgj has not provided any feedback on our rebuttal, despite our efforts to remind them during the discussion period. We hope that this situation will be considered when making the final decision.**

We are confident that BridgeVLA will make a meaningful contribution to the community and inspire further research in this field. We kindly request your consideration in supporting the acceptance of our paper.

Sincerely,

The Authors

---

### Decision · Program_Chairs · 2025-09-17

**Decision:**

Accept (poster)

**Comment:**

BridgeVLA is a robot manipulation model that uses a 2D VLM to process the information from 3D inputs (point clouds) in order to gain significant sample efficiency across a variety of manipulation tasks. The experiments (in the paper itself and in the rebuttal) show the method's advantage over the baselines and a high absolute performance in very low-data regimes.

The reviewers appreciated the data efficiency and generalization of the approach but pointed out the lack of heatmap-related ablations and comparisons to other VLAs as a weakness. The authors have addressed these by providing comparisons against SpatialVLA and pi_0 in the rebuttals.

The metareviewer finds that the authors have factually mitigated the reviewers' concerns but that reviewer Prgj's and J4oj's final justifications and scores are misaligned with the content of their discussions with the authors. Reviewer Prgj is giving the paper a firm reject because of a non-learnable component that doesn't appear critically important to the paper's contribution and despite their other concerns getting addressed. Reviewer J4oj is giving the submission a weak reject despite declaring the intention to raise the score, due to the paper's evaluations focusing on low-data regimes and supposedly missing in-depth explanations of BridgeVLA's advantage over pi_0. In the meantime, the authors explicitly state that BridgeVLA specifically focuses on low-data regimes, something that pi_0 isn't explicitly targeting and isn't designed to be suited for.

Overall, despite some weaknesses, this works appears to be above the NeurIPS acceptance bar.